# Arrow heads at Obi-Rakhmat (Uzbekistan) 80 ka ago?

Hugues Plisson[1]*, Alena V. Kharevich[2], Vladimir M. Kharevich[2], Pavel V. Chistiakov[2], Lydia V. Zotkina[2], Malvina Baumann[3,4], Eric Pubert[1], Ksenya A. Kolobova[2], Farhod A. Maksudov[5], Andrei I. Krivoshapkin[6]*

1 PACEA UMR5199, Université de Bordeaux, Pessac, France, 2 Institute of Archaeology and Ethnography SB RAS, Novosibirsk, Russian Federation, 3 Zoostan, IRL 2033, CNRS & KazNU, Almaty, Kazakhstan, 4 Traceolab, Université de Liège, Liège, Belgium, 5 National Center of Archeology, Tashkent, Uzbekistan, 6 APSACA, National Center of Archeology, Tashkent, Uzbekistan

* hugues.plisson@u-bordeaux.fr (HP); krivoshapkin@mail.ru (AIK)

## Abstract

Lithic weapon points occasionally found in Middle Palaeolithic Neanderthal sites are large and do not differ in size, shape or type from those used in other activities such as butchering or plant gathering. The presence in a same assemblage of various types of projectile armatures, some of which are microlithic and designed for this purpose, has only been documented in Modern Humans sites. Recent studies indicate that light projectile points, which would become a key element in Upper Palaeolithic lithic industries, were already present in its formative stages. However, they remain marginal in debates regarding the Middle to Upper Paleolithic transition. We present the initial findings of a traceological search for weapon heads in the oldest layers of the Obi-Rakhmat rock shelter in Uzbekistan, dating back around 80 ka. The lithic industry of this settlement is forming part of the Levantine Early Middle Paleolithic continuity but with several innovative traits. This site, located in the western foothills of the Tian Shan Mountains, northeastern Uzbekistan, has yielded throughout 10 meters of Pleistocene deposits covering 40,000 years a lithic industry characterized by the systematic production of blades (regular thick narrow blades from unipolar and bipolar sub-prismatic and narrow-faced cores, thin and wide blades from flat-faced Levallois-like cores) along with shorter pieces from convergent or centripetal Levallois cores, and bladelets from burin-cores and other small cores. Three types of projectile armature are identified over a selection of 20: retouched points, bladelets and more particularly unretouched triangular micropoints which had previously gone unnoticed due to their fragmentary state. According to the fundamental principles of hunting weapon design these micropoints are too narrow for having been fitted to anything other than arrow-like shafts. They resemble the armatures described in a pioneer settlement by Sapiens in the Rhône Valley, France, 25,000 years later.

**Data availability statement:** All relevant data are within the manuscript and its Supporting Information files.

**Funding:** Authors who did not received a specific funding: V.M.K., P.V.C., L.V.Z., E.P., F.A.M. H.P. was supported by the CNRS International Research Laboratory Artemir (https://www.cnrs.fr/fr) and by the French Institute for Central Asian Studies (https://ifeac.hypotheses.org/). The funders had no role in study design, data collection and analysis, decision to publish, or preparation of the manuscript. M.B. was supported by the CNRS International Research Laboratory ZooStan (https://www.cnrs.fr/fr). The funders had no role in study design, data collection and analysis, decision to publish, or preparation of the manuscript. A.V.K. was supported by the Russian Science Foundation (https://rscf.ru/en/), grant agreement # 22-18-00649. The funders had no role in study design, data collection and analysis, decision to publish, or preparation of the manuscript. K.A.K. was supported by the Project of the IAET SB RAS № FWZG-2025-0007 "The Application of Digital Technologies in the Analysis of Archaeological Data and the Reconstruction of the Ancient History". The funders had no role in study design, data collection and analysis, decision to publish, or preparation of the manuscript.

**Competing interests:** The authors have declared that no competing interests exist.

## Introduction

### Archaeological perspective

Instrumented hunting is a distinctive trait of the Homo genus. Given the impact of meat consumption on hominization, both cognitively and behaviourally [1], the search for archaeological evidence of past weaponry is of primary importance, with a particular attention to the oldest occurrences.

Increasing studies show that middle or small sized lithic points which are part of the typological characterisation of the Initial or Early Upper Palaeolithic assemblages were projectile heads [2–5], probably mechanically delivered [6–8]. They mark a technical break with the Middle Palaeolithic; from then on, projectile armatures will become the central structuring element of lithic industries (Bordes and Teyssandier, 2011). In spite of this, they remain marginal in the debates regarding the Middle-to Upper Palaeolithic *transition*. Already suspected of being present in Obi-Rakhmat sequence despite its age [9], light projectile points deserve attention.

We present here the first results of a search for weapon points in the oldest levels of the Obi-Rakhmat rock shelter in Uzbekistan at around 80 ka. The lithic industry of this settlement is forming part of the continuity of the Levantine Early Middle Paleolithic but with several innovative traits [10].

### Weapons in focus

Various criteria have been used to recognise prehistoric hunting weapons. The first is analogy with objects of comparable shape known from ethnographic records or from modern sporting or play practices. This is the case with javelins and throwing sticks from the ancient Palaeolithic [11]. More sophisticated approaches employ various acuteness indices, such as TCSA and TCSP [12–17], calculated for lithic points based on ethnographic and experimental data. However, these indices only represent theoretical potential [18,19]. As F. Bordes wrote: "*It could just as easily be argued that the sockets of bronze spearheads were used to cut rounds out of pie dough*" [« On pourrait tout aussi bien soutenir que les douilles des pointes de lance en bronze servaient à découper des ronds dans la pâte à tarte » [20]]. Furthermore, these indices fail to distinguish between simply tapered heads and those with cutting capacity [21], which makes a significant difference in real-life hunting, nor do they take into account the incidence on the penetration of the hafting device which depends on the morphology of the basal part. A more reliable basis is provided by functional clues. The most obvious evidence is when the tip of a perforating projectile is found stuck in a bone, though such finds are rare [22–31]. It is far more prevalent to find lithic or bone points that have sustained impact damage. However, this damage is subject to variation depending on the projectile design, the ballistic parameters and the impacted target. In contrast to the direct and invariable cause-and-effect relationship that characterises tools used for cutting, scraping, drilling, etc., where the same causes invariably produce the same effects, the identification of a projectile point or insert constitute an extrapolation based on the direction of the violent stress that resulted in the artefact fracture. It is evident that axial compressive stresses can be induced by factors other

than being at the tip of a spear or arrow, such as knapping accident [32–34] (S1 Fig: 2), certain types of shaping [35], hard butchering, use as a chisel, accidental dropping, and so forth. In certain instances, the differentiation is simple to make at the level of the artefact itself, in others it can prove more challenging if a series of criteria is not given full consideration. A compelling illustration is given by obsidian points from the Ethiopian rift dated to 279 ka years ago. These points were interpreted as javelin tips on the basis of their shape, apical removals and velocity-dependent microfracture features [36]. This last innovative criterion is physically relevant, but the presumed cause of the energy involved was likely not. A more extensive study on analogous assemblages, founded upon a technological analysis, posits that the recurrent apical removals on such pointed artifacts result from a rejuvenation by the lateral tranchet blow technique [37]. More commonly confusing are the minor damages that often occur at the extremity of pointed tools, which is their most exposed and fragile part. The negative of a tiny burin spall can turn a Levallois triangular flake into a projectile head [38, Fig 5].

## Obi-Rakhmat

The Obi-Rakhmat rock-shelter is located in the Paltau valley, at the south-western end of the Talassky Alatau range of the Tien Shan mountains, in northeastern Uzbekistan, 100 km of Tashkent. (N41°34'08.8" and E70°08'00.3") (Fig 1). Carved into the Palaeozoic limestone at an altitude of 1,250 m, it takes the form of a niche measuring 20 m in width and 9 m in length, with a southern orientation. Since the 1960s, several excavation campaigns have been conducted, initially by the Institute of History and Archaeology of the Uzbek Academy of Sciences [41], and later in collaboration with the Institute of Archaeology and Ethnography, Siberian Branch of the Russian Academy of Sciences [42,43].

The exposed stratigraphy (Fig 2) consists of 21 sedimentary levels spanning a depth of 10m [44], all of which contain archaeological material. The lithic industry, made from local silicified limestone, is homogeneous and characterized by the production of large blades from unipolar or bipolar and narrow-faced cores and bladelets from core-burins and bladelet cores of various morphology. These reduction strategies coexist alongside Levallois concept which is manifested by the presence of convergent or centripetal Levallois cores and flat-faced cores (Levallois-like). The typological tools include blades—often pointed and/or retouched— splintered pieces, burins, end- and side scrapers, denticulated, borers and retouched flakes. Notably, Levallois (mostly elongated) and Mousterian points are also present, the morphology of which is various and corresponds to the typology of retouched points in the Levantine Early Middle Palaeolithic [45,46]. This

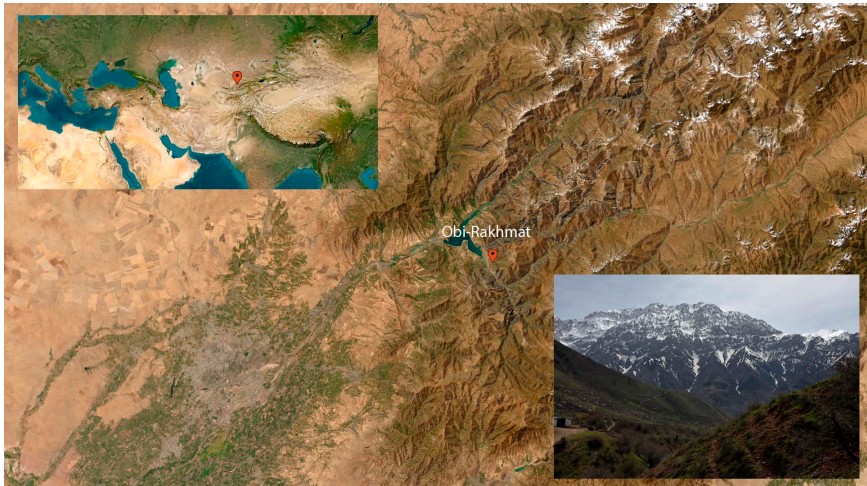

**Fig 1. Location of Obi-Rakhmat - N41°34'08.8" and E70°08'00.3", 1,250m asl – (basemaps courtesy of the U.S. Geological Survey/**https://usgs.gov**) and view from the site.**

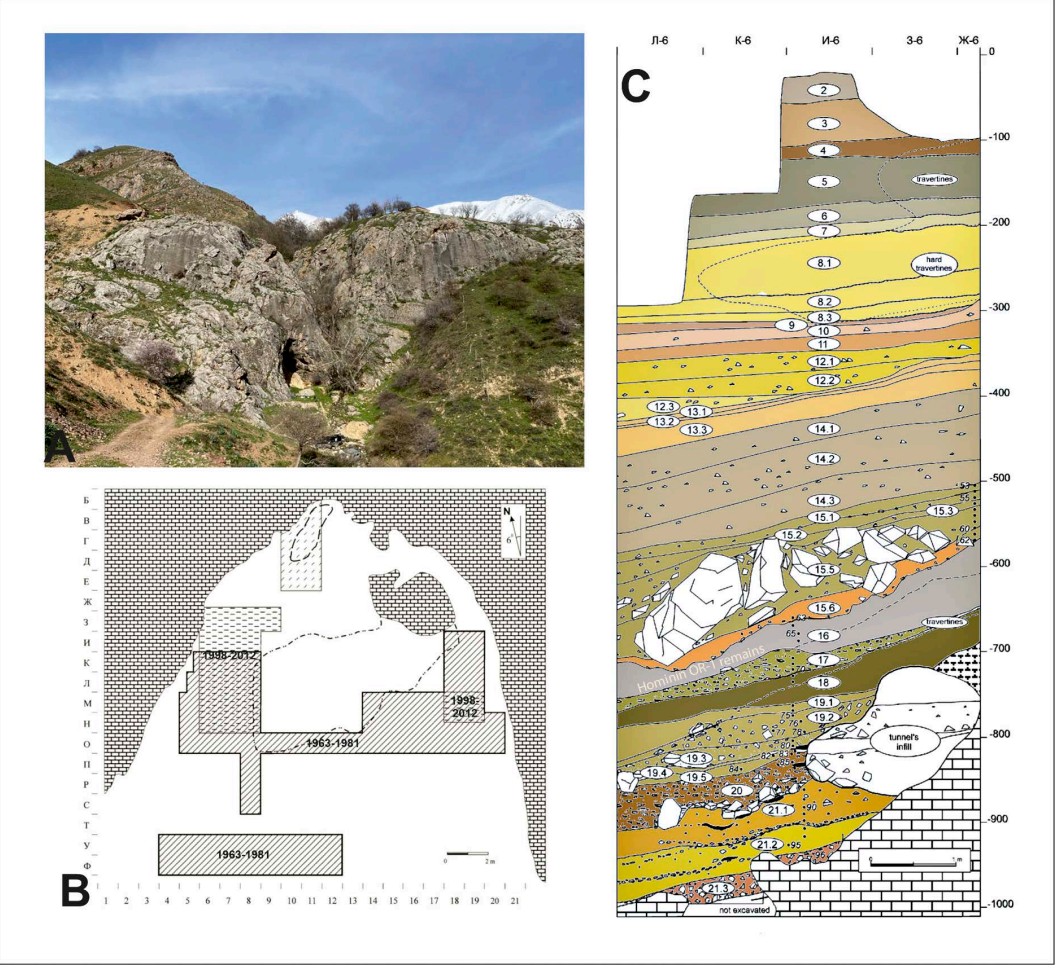

**Fig 2. Stratigraphy of Obi-Rakhmat rock shelter and map of the excavation.**

composition has led to comparisons between the Obi-Rakhmat lithic assemblage and the Late Middle Palaeolithic, Initial and Early Upper Palaeolithic blade industries [43,47] from Near East [48–51] and the Siberian Altai [52–55]. Current research considers the Obi-Rahmat industry exclusively within the Middle Palaeolithic framework [10]. Attempts to date the site have yielded heterogeneous results, as is often the case when applying different methods ([14]C AMS, U-series, ESR). However, they fix a chronological range from 90 ka for the deepest strata to 40 ka for the uppermost levels [56–58].

The faunal spectrum at Obi-Rakhmat is limited and shows little variation throughout the stratigraphy (Fig 2B). It is dominated by the Siberian ibex (*Capra sibirica*) and red deer (*Cervus elaphus*), with smaller contributions from wild boar (*Sus scrofra*), roe deer (*Capreolus capreolus*), golden jackal (*Canis aureus*), fox (*Vulpes vulpes*), marmot (*Marmota sp.*) and hare (*Lepus sp.*). This assemblage reflects a combination of steppe and forest environments [59,60], consistent with palynological data [61]. Remains of large carnivores, including cave lions, hyenas and bears, are rare. In 2003, human remains were discovered at the site: 6 left maxillary teeth and 121 skull fragments from a single juvenile individual (9–12 years old). While the dental morphology aligns more closely with Neanderthal populations, the mosaic of cranial morphological features prevents a definitive attribution, leaving open a classification as an archaic *Homo sapiens* [47,62–64]. Finally, elements of bone industry have been identified among the faunal remains [65,66].

## Materials and methods

The analysed sample comes from the collection of the 2001–2002 and 2007–2011 excavation campaigns led by A.I. Krivoshapkin, covering 20 m², which is currently under study. It comprises typological pieces and small triangular flakes that have been recovered from the bags of lithic debris from layers 20–21 stored at the National Center of Archeology in Tashkent, Uzbekistan. The initial sorting, which sought to identify impact damage, was done with the naked eye. Thereafter the selected specimens were examined with a stereoscopic microscope (Wild M1B/ x7, x14 magnification). Subsequently those exhibiting minimal erosion were analysed with a reflective optical microscope (Olympus BHM, bright field with DIC/ x50, x100, x200, x500 total magnification) to discern microscopic linear impact traces – MLIT [67].Single shots photomacrographs of the impact traces were captured in raw format using a Canon EOS 60D camera equipped with a Canon EF-S 60 mm f/2,8 Macro USM lens and photomicrographs using a Nikon D750 on the phototube of the microscope. Multi-focus shots were processed with Helicon Focus©..3D scanning was performed using a Solutionix D700 structured-light scanner, which makes it possible to create high-resolution non-textured 3D models of micropoints. A standard protocol for structured-light scanning of artifacts was followed [68]. Visualisation was carried out using Artifact-3D software [69]. The 3D PDFs of the supplementary data were produced in Acrobat Pro 9 from the STL files converted in MeshLab 2021.10. To allow each specialist to make their own judgement, the 3D model and photographs of each piece are provided in supplementary information.

As most of the artefacts are unsuitable for microscopic analysis due to the raw material and surface alteration, the determinations are essentially based on the morphology of the fractures, which depends on the ratio of compressive and flexural stresses that caused the material to break by buckling or percussion (Fig 3) , and lateral damage, as described in 40 years of publications [e.g., 2, 3,39,70–74].

Our personal experience is based on a corpus of over 500 flint points and barbs of various shapes used as arrow, dart and thrusted spear heads (Fig 3) on medium and large-sized mammals [75–82]. Notwithstanding their ease of use, gelatin targets, even when loaded with bone, cannot be recommended due to their inability to adequately reflect the stresses to which lithic points penetrating an animal's body were exposed and which determined their design (S1 Note) [21,83,84]. Additional experiments (S21 Fig, S2 Table) were conducted with the local silicified limestone to test the debitage scheme

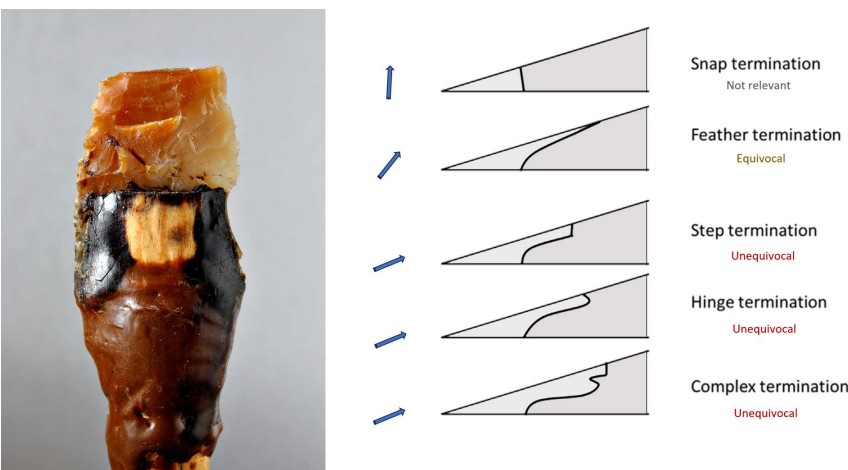

**Fig 3.  Terminations of bending fractures.** Left: experimental Solutrean leaf point used as dart head, broken at impact, with long step-terminating bending fracture (TFPS Collective Research Program). Right: profile of the different types of termination according to the resultant of compressive and bending forces [39,40] and their degree of relevance for the recognition of projectile points.

hypothesised on the basis of the shape of the micropoints and potential cores seen within the lithic assemblage. This step enabled the observation of the knapping accidents that could mimic impact damage. Twelve of the micropoints thus produced were fixed to 8 mm-diameter commercial wooden arrow shafts using a mixture 80/20% of bitumen (natural outcrops exist in the region) and pulverized charcoal, without binding (as this would be a technical absurdity on continuous cutting edges) for being shot with a modern laminated bow (36 lbs) on a complete uneviscerated carcass of a small ungulate hung in anatomical position in order to start documenting impact breakage and microscopic linear impact traces for the shape and raw material under study. Experiments demonstrated that silicified limestone exhibits comparable macroscopic fracture properties to flint, while MLIT are only observable on the most highly crystallized variants.

The Obi-Rakhmat set has been interpreted after the fundamental principles of projectile head design, as outlined in S1 Note, and compared with earlier and contemporary collections of projectile points, as well as with more recent assemblages featuring certain morphological similarities.

In order to reconstruct the production technologies of point and micro-point production, we performed an attributive analysis on cores, truncated-faceted pieces, and both points and micropoints. Within the Obi-Rakhmat *débitage* assemblage, points (>3 cm) and micro-points (≤3 cm) were distinguished. Layers 20–21 yielded a total of 194 point specimens. For micro-point analysis, 193 specimens were selected from the small flake category (1–3 cm). The study examined all cores from layers 20–21 (n = 96), a sample of points and micro-points exhibiting impact traces (n = 20), as well as those without impact traces (n = 326). Additionally, 36 truncated-faceted pieces from layers 20–21 were analyzed.

The following attributes were recorded for the cores: core type, morphology and measurements of the last negatives, number of negatives, angle between the striking platform and the flaking surface, striking platform type, and presence of overhang trimming. For the points and micropoints, the examined attributes included: type, morphology, scar pattern, overhang trimming, lip type, bulb type, striking platform type, striking platform width and thickness, and the angle between the striking platform and the ventral surface.

The reduction sequences of cores and truncated-faceted pieces were reconstructed through scar-pattern analysis [85,86]. Comparative analysis of selected metric parameters was conducted using a statistical approach, including the Mann–Whitney U test for pairwise comparisons.

Inclusivity in global research: Additional information regarding the ethical, cultural, and scientific considerations specific to inclusivity in global research is included in the Supporting Information (S1 Inclusivity). The archaeological materials analyzed in this study are curated by the National Center of Archaeology of Uzbekistan. This research was conducted under the Center's institutional protocols for scholarly access to collections, in collaboration with their research staff and as part of our ongoing academic partnership with the Institute of Archeology and Ethnography SB RAS (Novosibirsk, Russia). As this study exclusively employed non-destructive analytical methods and complied fully with the Center's established guidelines for collection use, no additional permits were required under current Uzbek cultural heritage legislation

## Results

### Impacted armatures

On the basis of the macroscopic criteria commonly used in publications and of our own experimental corpus, we have selected 20 pieces (Fig 4) that can be regarded as projectile armatures from the ongoing lithic inventory of the deepest levels (20–21) of Obi-Rakhmat.

Morphological recurrences and a diversity of distinctive traces are already apparent in this small preliminary sample. According to an agent-based modelling, it could even be considered high [87]. Projectile points and inserts are unusual artefacts in that they are not produced, used and discarded in the same place, and their rejection in the habitat differs depending on whether they are basal fragments brought back attached to the shafts or apical fragments returned in the carcasses [88]. Their density therefore depends on the type of site and the area excavated. In any case, the collection is

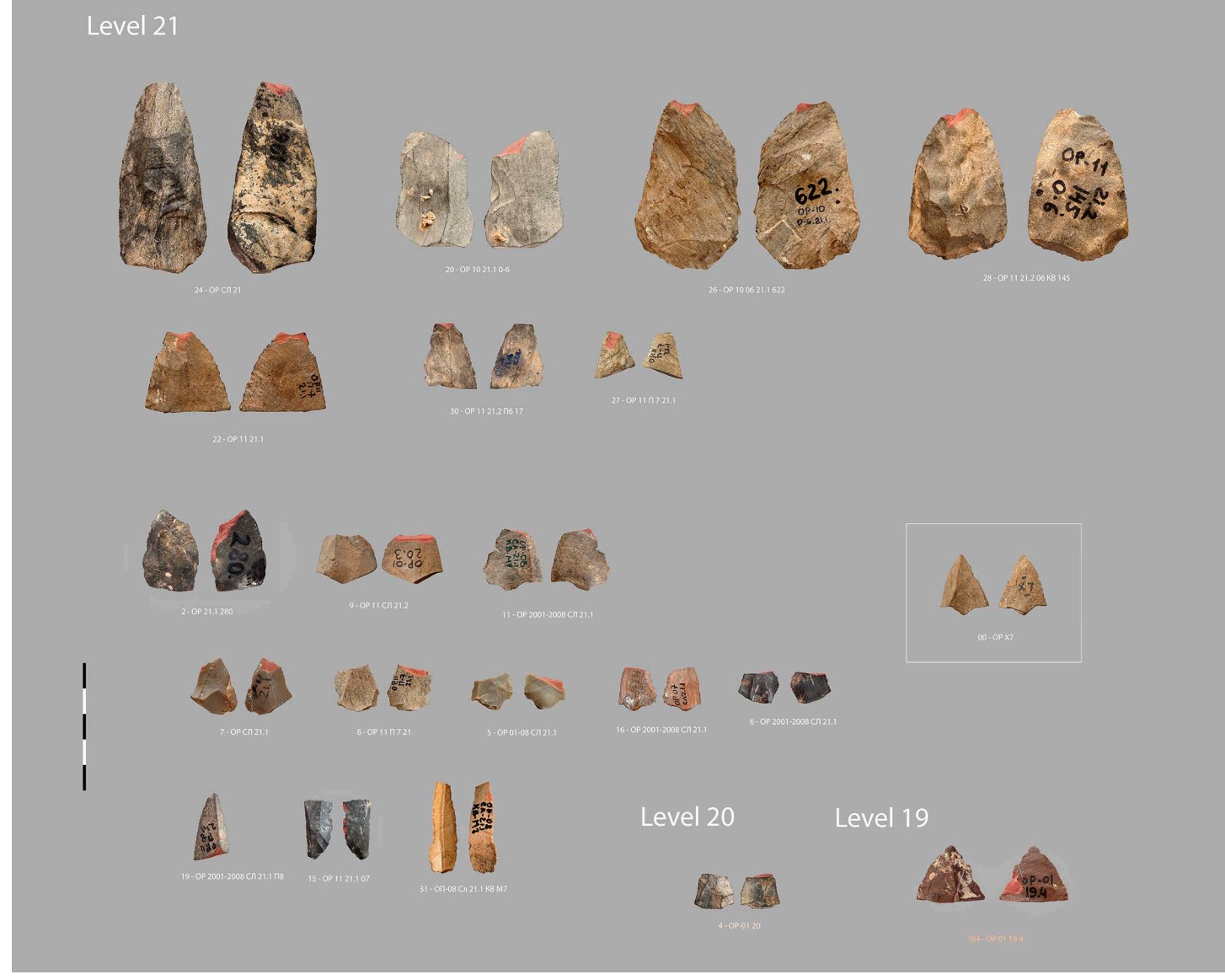

**Fig 4. Obi-Rakhmat: Impacted points and bladelets. The unbroken Levallois point in the white frame (00 – OP X7) illustrates what the ideal type of micropoint was likely to be.**

sufficient to distinguish 3 classes of projectile armature: more or less massive points (between 25 and 35 grams for almost complete specimens), micropoints (between 1 and 4,5 grams but broken) and bladelets (Fig 5).

The large points (Fig 6) are represented by 2 almost complete retouched points (Fig 6: 28, S1 Fig, S1 File, S2 Fig, S2 File) (38 mm and 41 mm wide) and an apical half (Fig 6: 22, S3 Fig, S3 File), all 3 crushed at the tip by an axial impact. An apical fragment of retouched point from layer 19 is similar, having been badly chipped by a tangential impact to its right edge along the same axis (Fig 6: 104, S4 Fig, S4 File). The proximal thinning on both sides of the complete specimens suggests a hafting layout. A thicker but more elongated point (Fig 6: 24, S5 File, S5 File) (30 mm wide) has an apical fracture with a wavy feather termination on the lower side from the same type of compressive stress.

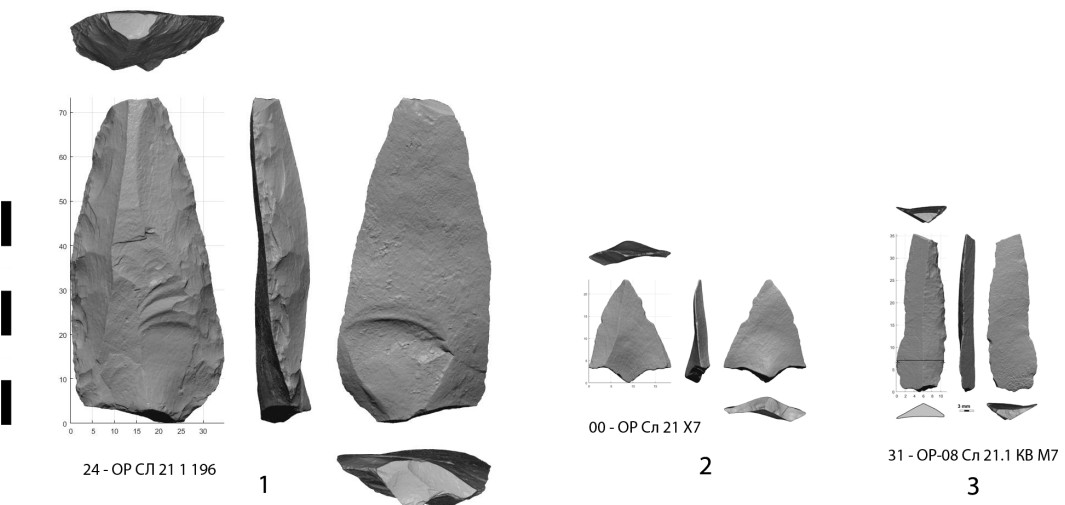

**Fig 5. Three types of lithic weapon armature identified in layers 20 and 21 of Obi-Rakhmat: 1 – Large retouched point, 2 – micropoint (Levallois) and 3 – bladelet.**

24 - ОР СЛ 21 1 196　　1

00 - ОР Сл 21 X7　　2

31 - ОР-08 Сл 21.1 КВ М7　　3

One large proximal fragment (Fig 6: 20, S6 Fig, S6 File) which may be from the same type of point as the previous one, also has a fracture attributable to its use as spear tip, but it is lateral, which can only occur under the leverage of a long shaft.

Two apical fragments from more slender retouched points complete the set. One has a bending fracture with an atypical feather termination but here slightly twisted (S7 Fig, S7 File), while the other has a long spin-off fracture (S8 Fig, S8 File).

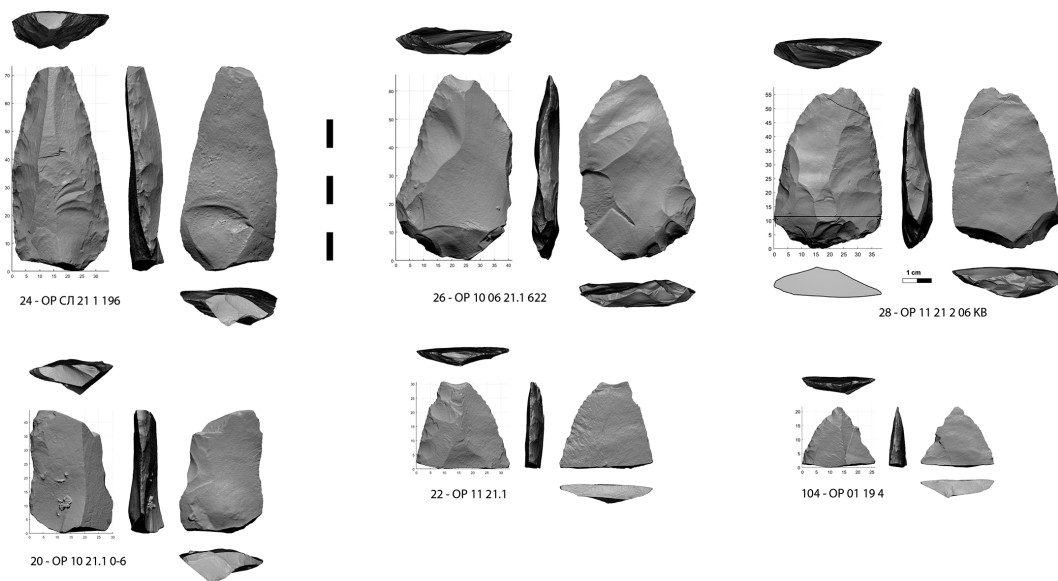

24 - ОР СЛ 21 1 196

26 - ОР 10 06 21.1 622

28 - ОР 11 21 2 06 КВ

20 - ОР 10 21.1 0-6

22 - ОР 11 21.1

104 - ОР 01 19 4

**Fig 6. Large retouched points crushed or broken at impact.**

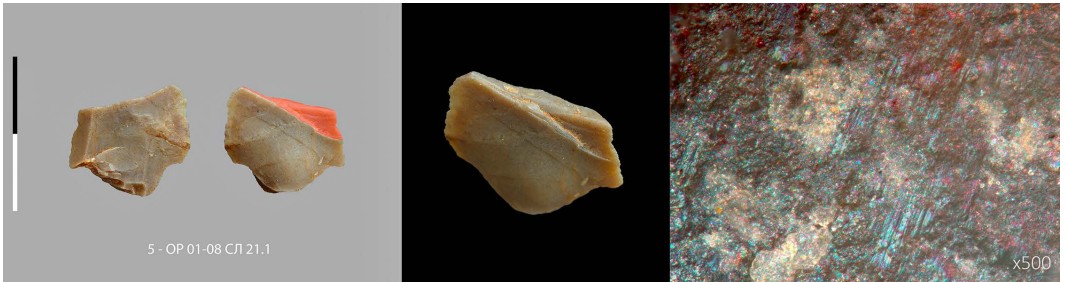

00 - OP Сл 21 X7

5 - OP 01-08 СЛ 21 1

4 - OP-01 20

6 - OP 2001-2008 СЛ 21.1

7 - OP СЛ 21.11

8 - OP 11 П 7 21

16 - OP 2001-2008 СЛ 21.1

2 - OP 21.1 280

5 mm

9 - OP 01 СЛ20.3

11 - OP 2001-2008 СЛ 21.1

**Fig 7. Micropoints broken or crushed at impact.**

5 - OP 01-08 СЛ 21.1

x500

**Fig 8. Micropoint n° 7 (5 – OP 01-08 СЛ 21.1) with macroscopic and microscopic impact traces.** Centimetric scale.

The second category consist of 9 unretouched micropoints and 1 retouched micropoint (Fig 7, S9–16 Figs, S9–16 Files), produced in a variety of ways, from simple triangular flakes to typical Levallois points. Two specimens have both fracture from longitudinal stress and MLIT (Figs 8,9). The average width is 18.2 mm (minimum 15,0 – maximum 23,7 mm). The average weight of the fractured specimens is 1.4 grams (minimum 0.7 g – maximum 2.5 g) (S3 Table). An unbroken Levallois micropoint (Fig 7: 00, S17 Fig, S17 File) measures 21.8 mm long x 17.3 mm wide and weights 1.1 grams; its edge damage is atypical (trampling or crushing). There is no basal shaping despite prominent bulbs. The location on the lower side of the fracture hinge for 6 specimens, half of which have elongated endings, whereas a flat surface is not conducive to their extension, suggests a mounting on the shafts that does not compensate for the prominence of the bulb, i.e., a mounting that is not perfectly in line with the axis (which is not optimal for the wake drag coefficient). The shafts were probably made of wood only, as no potential intermediate nor apical bone elements were found [65].

The 3rd category (Fig 10) is the least represented here because it was not included in our sorting criteria, which focused on axial armatures. Four incidental findings came from this sorting. These are raw bladelets. Without retouch or MLIT it is not possible to distinguish between those accidentally broken during knapping [32,34] and those broken by an axial use. They were therefore excluded from the current inventory. Two other bladelets were selected on the basis of the following criteria: one (Fig 10: 19, S18 Fig, S18 File), whose breakage by bending is equivocal, has a very discreet retouch along one edge caused by pressure that appears to be postdating the fracture; the second sample (Fig 10: 15, S19 Fig, S19 File), probably burnt from dehafting, has been crushed by a tangential contact with a hard material, which

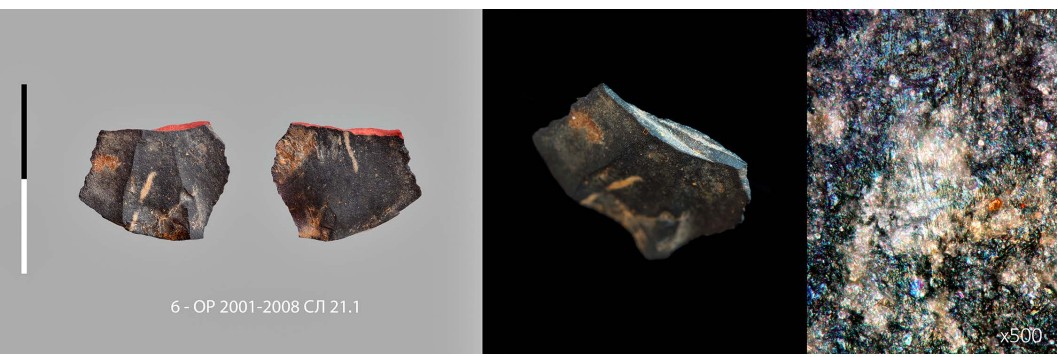

**Fig 9. Micropoint n° 8 (6 – OP 2001-2008 СЛ 21.1) with macroscopic and microscopic impact traces.** Centimetric scale.

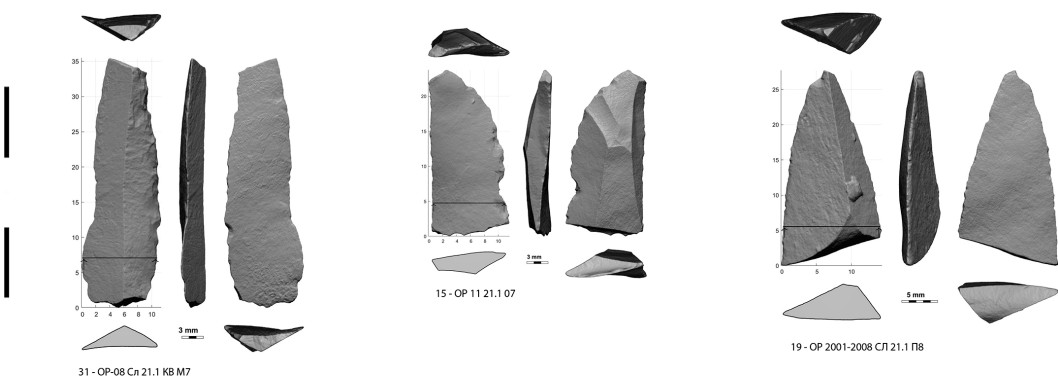

**Fig 10. Bladelets crushed of broken at impact.**

is typical of a projectile lateral insert. A more discreet crushing is also observed on a genuine backed bladelet (Fig 10: 31, S20 Fig, S20 File).

The set, although small, indicate the use of bladelets in the design of hunting weapons at Obi-Rakhmat, 80 ka ago. We could imagine that they complemented the micropoints, to extend the length of the cutting edge, thus forming part of a more robust projectile head compatible with dart foreshafts, but nothing in the design of the micropoints would ensure a continuity to prevent the bladelets from being pulled out at the penetration. Recent archaeological examples show the extreme attention paid to the continuity of lithic inserts [89]. A larger sample and therefore specific research would be required to understand the function(s) of the bladelets, which are a notable element of the lithic production at Obi-Rakhmat.

### Technological characterization of production

**Point production.** The primary reduction system in Obi-Rakhmat layers 20–21 exhibits significant diversity. This phenomenon is evident in the morphology of the cores themselves, as well as in the typology of the technical flakes and blanks. The industry places significant emphasis on the Levallois techniques, including Levallois for points and centripetal Levallois, as well as blade production through the exploitation of flat faced, sub-prismatic, and narrow-faced cores. In addition, bladelets were also produced using various core types.

The diagnostic blanks include blades, bladelets, and flakes in nearly equal proportions. Depending on the layer, points constitute up to 10% of all flakes (S1 Table). Notably, according to the tool assemblage, pointed blank morphology was of particular importance (Fig 11). A significant proportion of the tools in layers 20–21 consist of various types of retouched pointed implements (S1 Table).

Given the broad diversity of reduction sequences employed in the assemblage and the high proportion of retouched points among the tools, we propose that blanks for pointed tools were not limited to predetermined Levallois points but also included suitable technical flakes with pointed morphology. This observation is applicable to both pointed tools in general and the sample of impact-fractured points.

In the assemblage of layers 20–21, Levallois point cores (Fig 12A) constitute 5–30% of the total (S1 Table). Scar-pattern analysis indicates that pointed blanks were produced both as predetermined products and technical spalls from blade cores. The collection includes flat-faced unidirectional (Fig 12: 2) and bidirectional (Fig 12: 3) cores, as well as a considerable proportion of narrow-faced cores (Fig 12: 4-5), which frequently exhibit pointed negatives. Another techno-logical method (Fig 12D), which is extensively represented in the collection, is associated with subprismatic unidirectional (Fig 12: 6) and bidirectional (Fig 12: 7) cores. Among the subprismatic cores, there are asymmetrical (semi-rotated) cores (Fig 12: 8) that combine exploitation of both large and narrow core faces.

**Micropoint and bladelet production.** While micropoint production has not been a focus of previous studies in the Obi-Rakhmat assemblage, the identification of impact-damaged specimens in the use-wear sample now necessitates an examination of this reduction sequence. Initial investigations do not reveal a clearly defined *chaîne opératoire* dedicated exclusively to micropoint production. We have systematically examined all cores for the presence of small triangular negatives corresponding to the dimensions of impact-crushed micropoints – with particular attention to the characteristics of the proximal zone.

As with the larger blanks in the Obirakhmatian industry, we observe considerable diversity in the cores aimed at producing small blanks – bladelets and small flakes (Fig 13). Based on the morphology of the débitage itself, the dimensions and morphology of the primary and secondary negatives, we conclude that micropoints could have been produced by several reduction sequences. One of the predominant reduction sequences appears to have involved: small Levallois cores present in the collection (Fig 13A); and small flat faced cores-on-flakes (Fig 13B) that yielded small flakes and bladelets

While truncated-faceted pieces (Fig 14) have been interpreted as micro-point cores in other contexts [6,90], we argue that within the Obi-Rakhmat technocomplex, most such artifacts functioned mainly as tools [46].

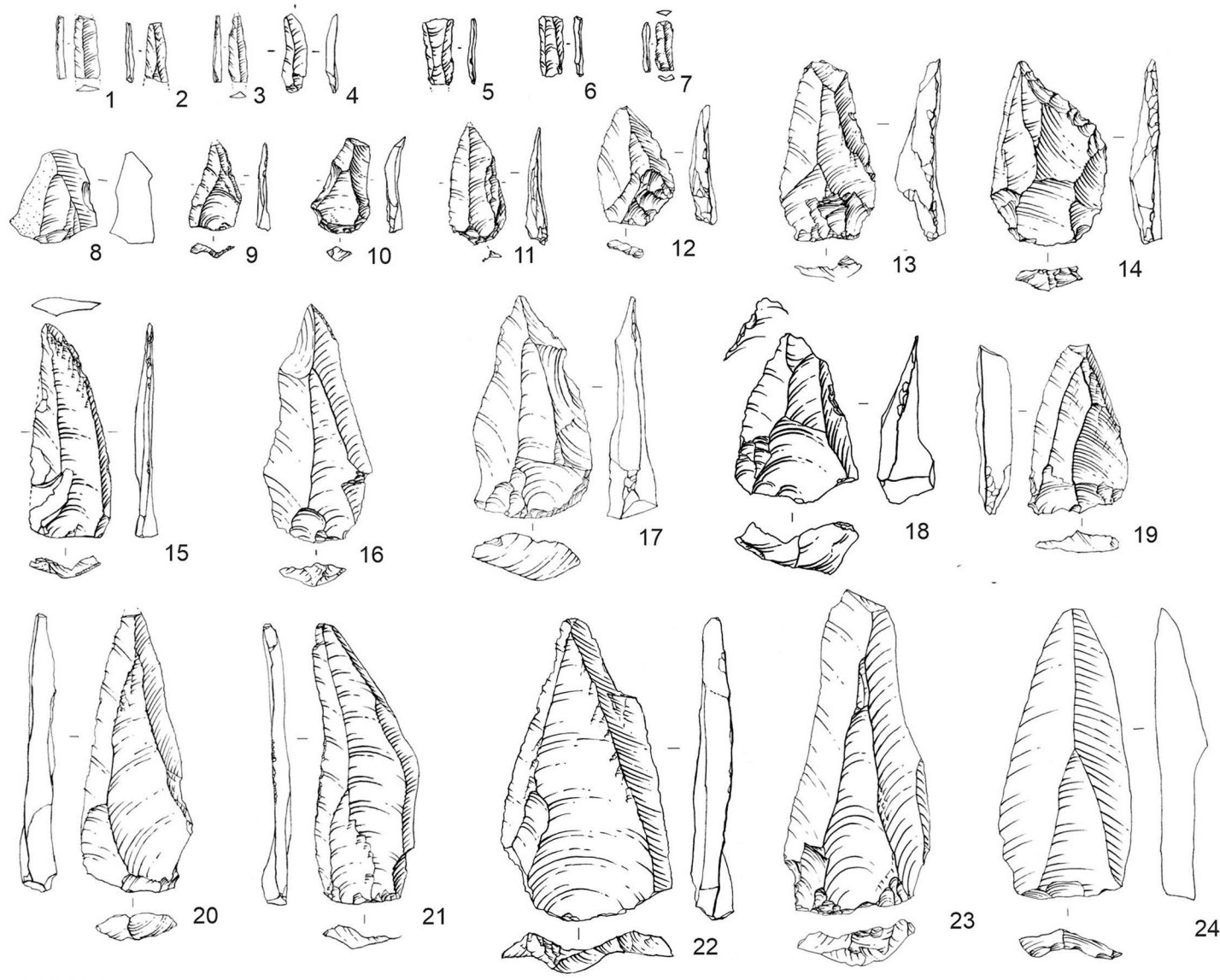

**Fig 11. Convergent spalls from layers 20-21. 1-6 Bladelets; 7-10, 17-18, 21-23 Levallois points; 11-13, 16 Axial points; 14-15, 19-20 Elongated points.**

This interpretation is supported by several key observations: (1) their highly standardized morphology emphasizes deliberate working edge formation rather than core reduction features; (2) they rarely exhibit the triangular negatives characteristic of micropoint production; and (3) although they often possess prepared striking platforms, these show no evidence of flake removals. Crucially, metric analysis reveals that truncated-faceted pieces differ significantly from formal cores in striking platform angles (Fig 15). Moreover, the platform angles of these truncated-faceted pieces show no correlation with the residual angles observed on the analyzed points, in marked contrast to all other core types in the assemblage. This discrepancy (confirmed by Mann-Whitney U tests, $p < 0.01$) further demonstrates their technological separation

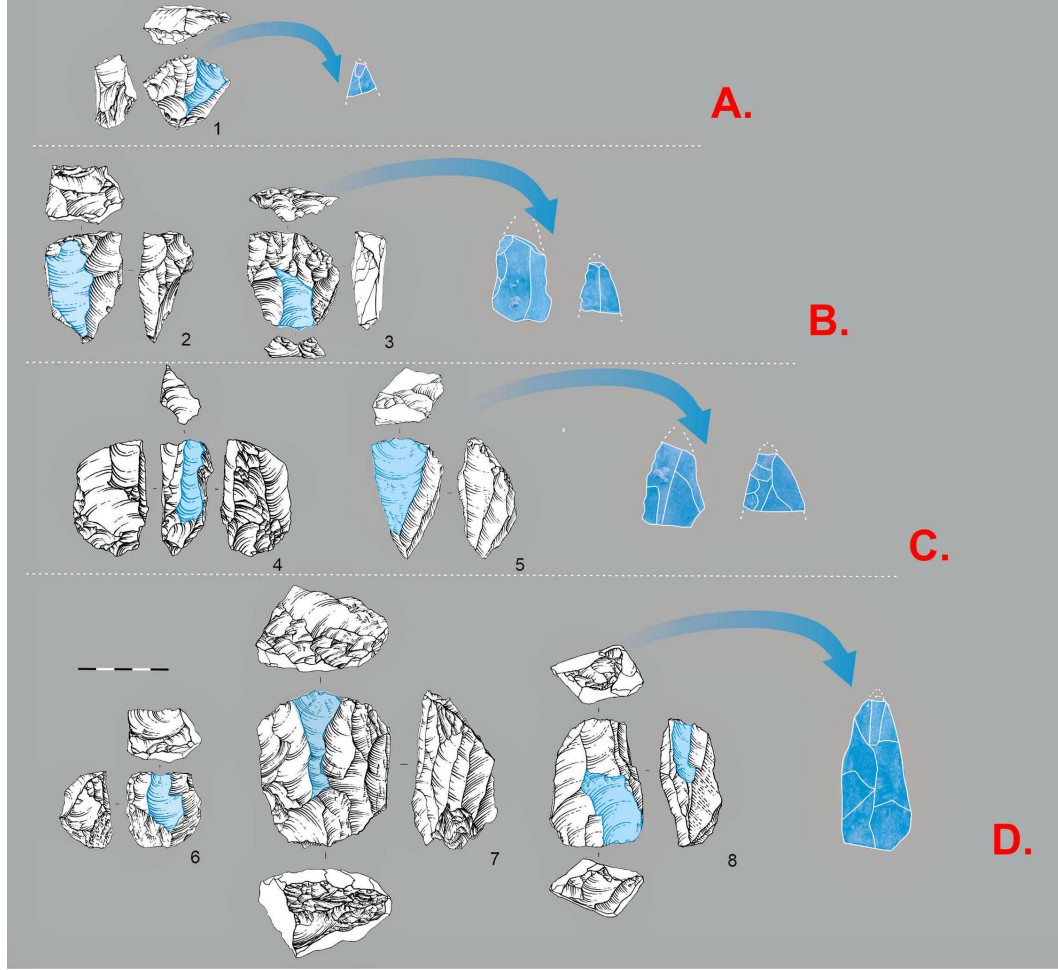

**Fig 12. Technological chains of point and elongated blade production.** A. 1 Levallois point core. B Flat faced cores: 2 Flat faced, unidirectional; 3 Flat faced, bidirectional. C. Narrow-faced cores: 4-5 Narrow-faced unidirectional. D Sub-prismatic cores: 6 Sub-prismatic unidirectional; 7 Sub-prismatic bidirectional; 8 Sub-prismatic asymmetrical (*Semi-tournant*).

from systematic core reduction activities. Fig 15 presents boxplots showing: (1) angles between flaking surfaces and striking platforms of different core types, (2) angles of truncated-facetted pieces, and (3) striking platform angles preserved after point detachment (measured from points in our sample).

Scar-pattern analysis showed that flakes with the required triangular morphology could also be produced during bladelet production. A developed bladelet production is widely represented in the lithic industry (S1 Table). The assemblage of lower layers is comprised of 20–30% bladelets (S1 Table), with the proportion of bladelet cores reaching 50% (S1 Table). The bladelet cores used in the Obi-Rakhmat industry exhibit a high degree of diversity, with the presence of carinated (Fig 13E), narrow-faced (Fig 13C: 5), burin-cores (Fig 13C: 4), and sub-prismatic (Fig 13D) cores. The utilization of all these technological chains yielded micro points, as well as pointed bladelets that exhibit a morphological resemblance to the archaeological specimens that were selected as projectile armatures.

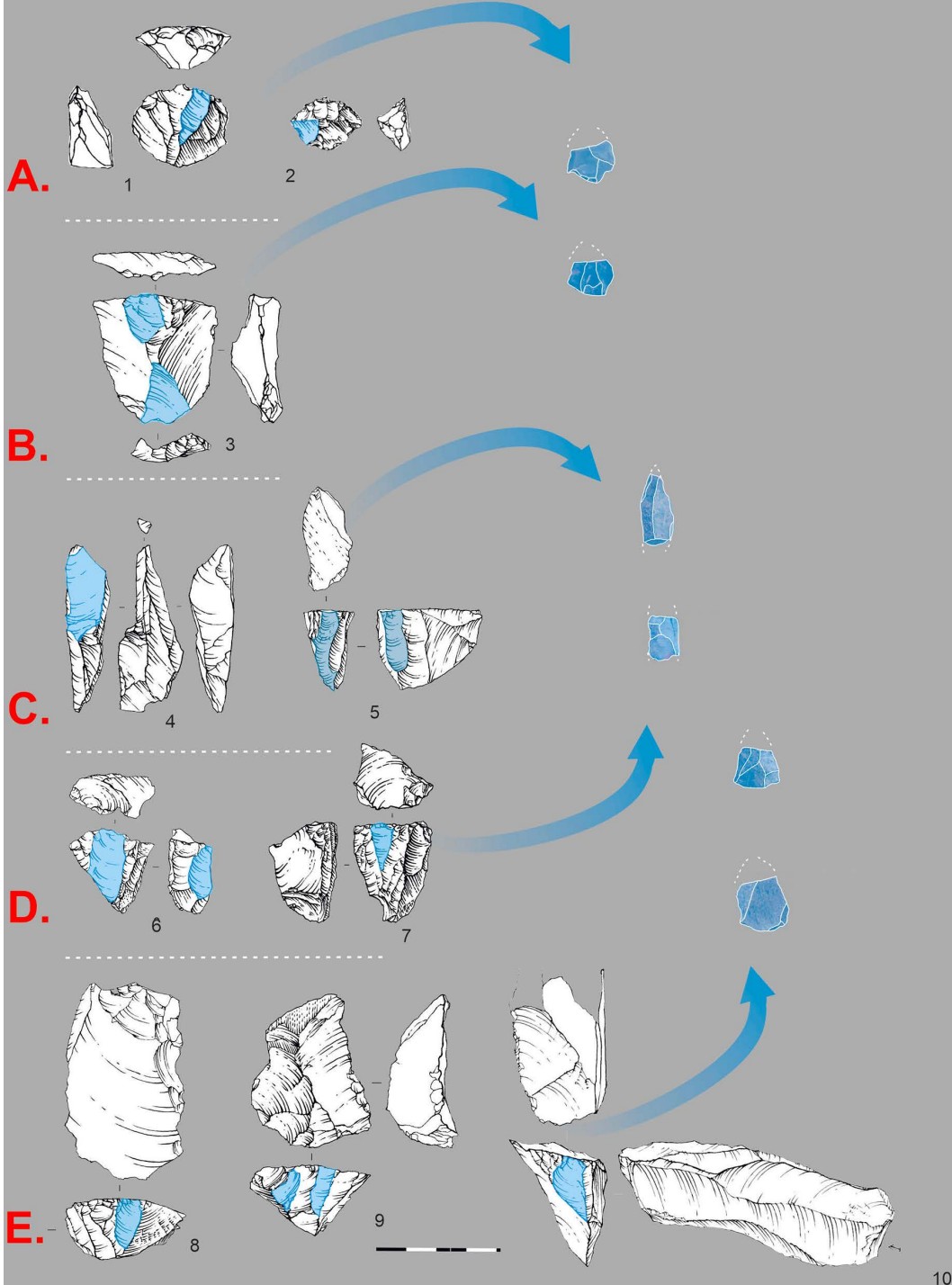

**Fig 13. Technological chains of micro-point and bladelet production.** A. 1-2 Small Levallois point cores. B: 3 Flat faced, bidirectional core on flake. C: 4 Burin-cores, 5 Narrow-faced core. D: 6-7 Sub-prismatic, unidirectional cores. E: 8-10 Carinated cores.

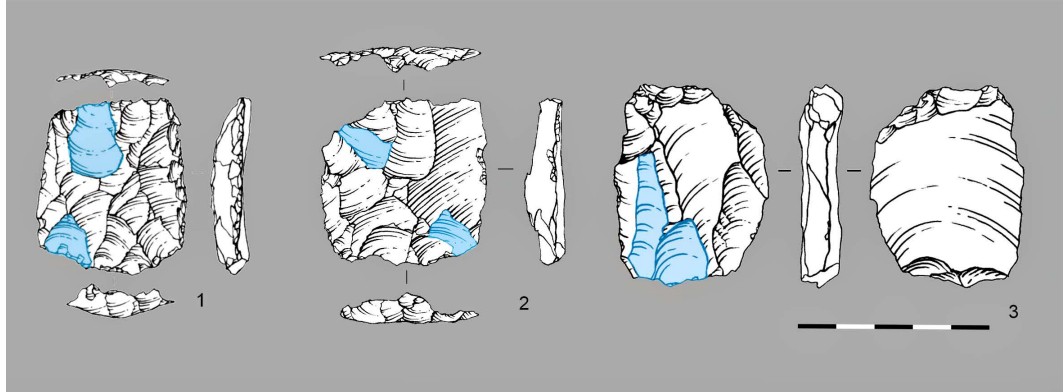

**Fig 14. Truncated faceted pieces from layers 20-21: 1-3.**

## Discussion

### Spear or arrow?

The question that usually arises with armatures is the type of weapon on which they are mounted. There are 6 different types of piercing weapons: daggers, hand-held stakes, hand-thrown spears, spears and darts thrown with a spearthrower, arrows shot with a bow and blowpipe-fired darts. With the exception of the dagger, which is an ancillary weapon for hunting, this classification according to distance of use also corresponds to the weight (and robustness) of the weapons in relation to the kinetic energy involved in their use, from the heaviest to the lightest. At the two ends of this range, the critical parameters are reversed: on the one hand, robustness is essential, while penetration is guaranteed in any case by the high power developed; on the other hand, the low impact power makes the sharpness of the projectile tip a determining factor. The design of the armatures will therefore depend on these constraints, in terms of shape and size. Numerous studies have attempted to find discriminating traceological criteria for distinguishing throwing techniques on the basis of the lithic points damage. They converge on the conclusion that the degree of damage (position of the fracture, length of the termination, number of fragments, etc.) is proportional to the energy involved [80,91,92], i.e., javelin points are more broken than arrowheads (all other things being equal), but none has fixed a threshold because there are too many parameters involved. Even admitting that the experimental conditions could be strictly similar to the prehistoric ones (weapon characteristics, type of mounting, method of delivery, shooting distance, hardness of the surrounding ground, ambient temperature, game, etc.) – which is an illusion – a quantitative model would only be discriminating in a homogeneous archaeological assemblage in terms of all the parameters considered. The accumulation of points of the same type but adapted to both arrows and darts, or shot in summer and winter [93], is likely to render it inapplicable.

The only discriminating qualitative criterion between arrows and spear heads, for ambivalent points (e.g., shouldered points), is the breakage by lateral flexion (from one edge to the other, not transversally) induced by the lever arm of a long shaft [3].

Recently, a new set of criteria has been highlighted, based on the ratio between the bending and compressive components of the point fracture according to the ballistic trajectory of the projectile. However, the distinction is currently only relevant between hand-thrown javelins on the one side and arrows or darts on the other [74,94].

In any case, the current series of Obi-Rakhmat is far too small to apply statistical model based on traceological criteria. In order to hypothesize the type of weapon on which the points were mounted, we must consider the technological parameters of their functioning. A point 1.5 cm wide and weighing between 1 and 2 grams does not meet the same needs as a point 4 cm wide and weighing 30 grams.

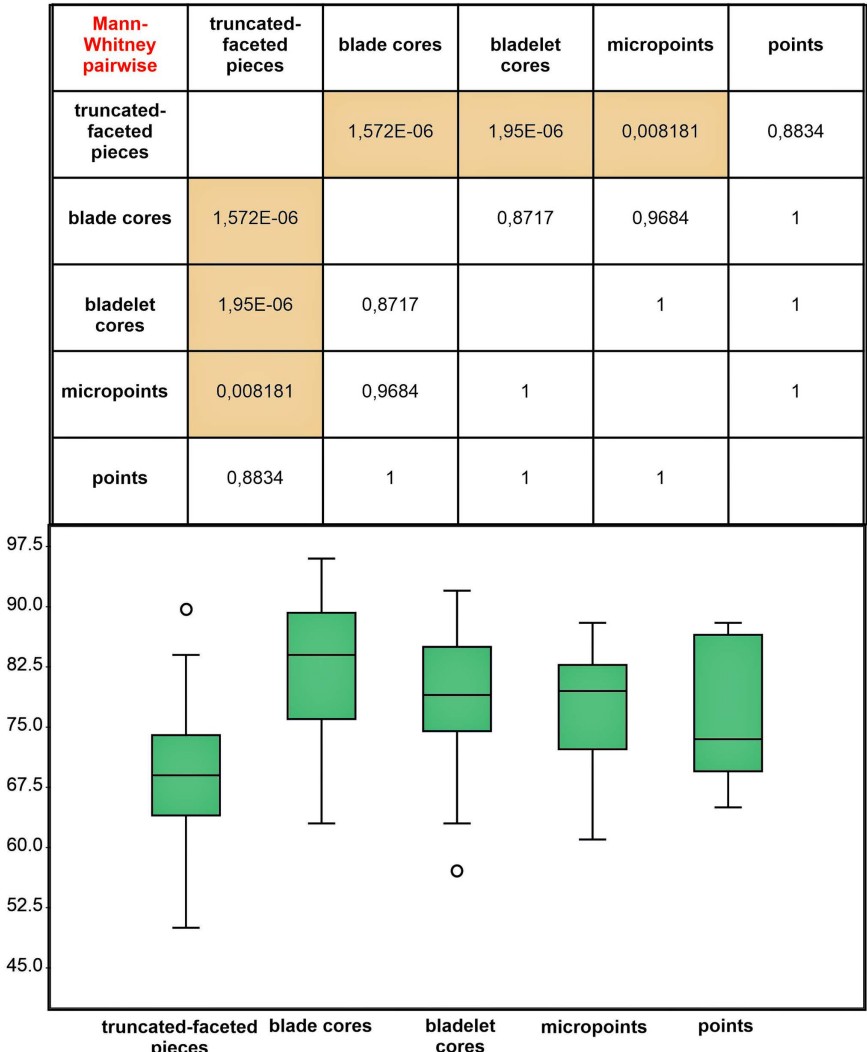

| Mann-Whitney pairwise | truncated-faceted pieces | blade cores | bladelet cores | micropoints | points |
|---|---|---|---|---|---|
| truncated-faceted pieces | | 1,572E-06 | 1,95E-06 | 0,008181 | 0,8834 |
| blade cores | 1,572E-06 | | 0,8717 | 0,9684 | 1 |
| bladelet cores | 1,95E-06 | 0,8717 | | 1 | 1 |
| micropoints | 0,008181 | 0,9684 | 1 | | 1 |
| points | 0,8834 | 1 | 1 | 1 | |

**Fig 15. Comparative analysis of the angles between the flaking surface and the striking platform of cores, as well as the dorsal angle of points.** Residual point angles were calculated using the formula: 180° minus the angle between the striking platform and the ventral surface. Raw data are in S4 Table.

In the category of large points, between tip crushed samples (Fig 6: 22, 28, S1 Fig) and a heavily laterally broken one (Fig 5: 20), the range of energy involved is too wide for identifying a particular weapon, while their size could fit dart and spear. We can only note that the points with the sharpest edges could have been mounted as daggers, a type of weapon that is rarely considered in the studies [82]. Unfortunately, their poor state of preservation does not allow a detailed traceological analysis. Conversely, the narrow but thick point n°24 (Fig 6: 24), with less sharp edges, seems to have been designed for a heavy dart or spear such as the broken one n°20 (Fig 6: 20).

The category of small points is more revealing. Such small points were not designed to withstand violent impacts, their triangular shape made it impossible to bind them to the shafts, only to glue them, and these shafts could only be significantly smaller in diameter than their maximum cutting width [95]. It is important to remember that the role of the cutting armature is to tear the skin of the prey to open the way for the shaft, so that the skin does not tighten around it and reduce

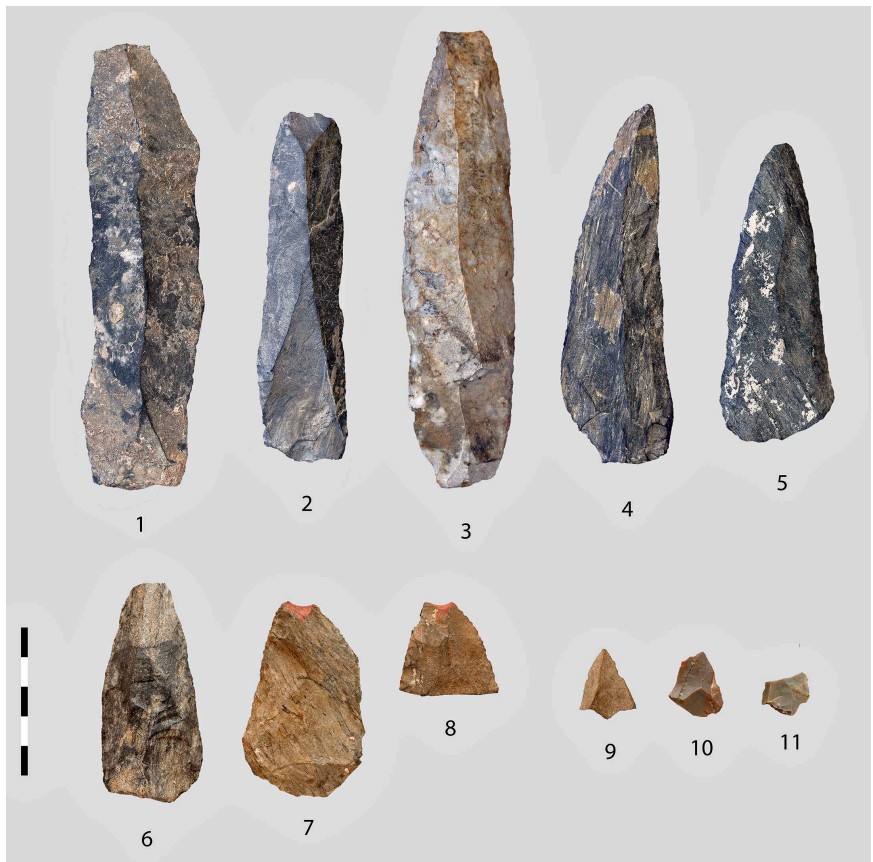

**Fig 16. Micropoints alongside structuring products of the lithic industry.** 1-5 blades and elongated points from blade cores, 6-8 Large retouched weapons heads, 9-11 Micropoints. Centimetric scale.

penetration. This is all the more important as the projectile has low momentum, which is the case for an arrow shot with a primitive bow. Contrary to what has been endlessly debated by archaeologists, it is not the weight of the arrowhead that first matters but its ability to penetrate, which is directly related to its width and sharpness for a given bow draw-weight [83,96–98].

Inventories of archaeological and ethnographic arrows, and even contemporary recreational and sport archery, show that their shafts vary very little from a diameter of 7–8 mm [95,99–103], depending on the power of the bow. Contrary to what Klaric et al. claim [104], this is not a "rule", i.e., a cultural arbitrariness, but the consequence of physical principles, i.e., a transcultural invariance. This is what A. Leroi-Gourhan called "trends", as opposed to the "degrees of fact" [105].

This has to do with the flexibility (*spine*) of the arrow [73 Supplementary file2 video], which, if too great, will absorb some of the energy at launch and at penetration and induce too much oscillation in flight and at impact, or, if too weak, will not pass around the grip of the bow and will deviate laterally, in both cases altering the accuracy of the shot [83,95,106]. The dart's spine [[74] Supplementary file3 video, 107] needs to be calibrated just as precisely, but according to the spear-thrower operating principle to which it contributes, with the same consequences for the efficiency of the shot [108]. One of the parameters used for adjustment it is the resistance to acceleration induced by the mass of the tip [95,109].

Last but not least, the resistance to penetration increases with the diameter of the shaft due to the greater surface area exposed to the tissues, resulting in a higher drag factor [83].

The difference in the principle of energy delivery between the bow and the spear-thrower, with an inverse relationship between velocity and mass, results in projectiles whose characteristics are not interchangeable and whose impact force is not the same (higher with darts). As Hughes wrote "*Spine and size conformance (matching) between the projectile and the propulsive device are critical to insure accuracy and an efficient transfer of energy* [[110] Supplementary file3 video, " [95].

This explains the bimodal distribution distinguishing darts and arrows, according to head width and shaft diameter, reported by Marsh et al. [100 Fig 4] from the measurements of 168 hafted points or shafts across the American continent. The chosen geographical area, with its rich archaeological and ethnographic corpus, also offers the advantage of having known only one type of bow before the Hispanic period – the simple or self-bow – and the widespread use of composite arrows and darts made from a main shaft and a removable foreshaft. This technical solution is not exclusive to the American continent, as evidenced by the complete arrows from Egyptian Predynastic and Dynastic tombs (V and IVth millennia BC) whose lithic and bone armatures have typological equivalents in the Epipaleolithic and even the Late Stone Age [111]. It allows the use of shorter natural shafts than when making one-piece javelins, and the interchangeability of projectile ends, but with the disadvantage of a lesser penetration (no deeper than the connection) and greater breakability. The foreshaft reduces the dimensional gap between the arrow and the dart head, by allowing smaller points on composite darts than on single-piece ones. However, the dart foreshaft must be more robust in order to withstand greater stress than the arrow (greater impact force with more bending component [74]), which means a larger diameter.

To cut the skin widely enough for reducing the resistance to shaft penetration, Hughes calculates that the maximum circumference of arrow and dart tips should be 140–150% of the shaft diameter [95]. The complete micropoint (Fig 4: 00, Fig 7: 00, S17 Fig), whose maximum thickness is not regular, with a peak at 1.5 mm on one ridge and another at 3 mm on the second, and one cutting edge at 1 mm, is frontally closer to a furtive airplane than to the simple geometric shape to which the TCSP calculation formula corresponds. The main issue with the micropoints is in the calculation of their frontal outline, since the greatest thickness is frequently at the level of the bulb, which is in the continuation of the shaft and therefore, like the shaft, depends on the morphology of the cutting edges beyond its diameter for the opening of its way. Nonetheless, even if we use the maximum thickness value, the calculation gives a potential shaft diameter of 7.5–8 mm, i.e., (23,6–25,1 mm perimeter) that of an arrow. Because of their quasi-equilateral outline, the triangular micropoints of Obi-Rakhmat have geometrically a lower cutting efficiency (*mechanical advantage*) than the Paleo-Indian points of thin elliptical cross-section and much longer cutting edges from which this index has been extrapolated [95,112]. As a result, the corresponding theoretical shaft diameter should be regarded as a maximum.

In order to refute the presence of arrowheads in contexts too ancient for considering the existence of bow and arrow technology, Klaric et al. have recently proposed the alternative hypothesis of reduced weapons for children with which the criterion of efficiency becomes secondary [104]. On the basis of ethnographic data from three continents, they distinguish two archaeologically demonstrable cases: simplified lithic or bone points expeditiously made by adults or clumsy ones made by children. It's true that the place of children in prehistoric societies is not considered a research topic in itself, even though its anthropological implications would be fundamental, particularly for the Middle or Early Palaeolithic, but the criteria proposed do not fit in with Obi-Rakhmat's industry. The micropoints, there, are not reduced replicas of the larger retouched ones nor do they result from a same debitage scheme (Fig 16). Unlike the ethnographic and archaeological examples mentioned by Klaric et al., these are not shaped points but knapped ones, the experimental reproduction of which requires a real know-how. This appear evident when examining the butt of the archaeological specimens. To assume that children learned lithic knapping by using their own solutions, different from those practiced by adults, is not the kind of parsimonious hypothesis advocated by Klaric et al. While there is currently no experimental model to show that children can throw small arrow-sized spear or dart with sufficient force to be effective in hunting condition (the MLITs of 2 micropoints indicate that they penetrated soft matter before breaking against something hard, which is usually the structure of an animal or human target).

Unlike bifacial points, whose final width may be the result of several stages of rejuvenation, the width of unretouched knapped points when discarded is that of their intended design. For the time being, the simplest hypothesis is that the production of the micropoints at Obi-Rakhmat was a response to a specific need for small, lightweight sharp armatures.

A technical alternative to bow and arrow for such small projectile points, but which to our knowledge has not been experimentally tested, could be that of a dart armature shot with a blowgun. However, the ethnographic examples documented in South-East Asia and on the American continent [113], originated from a core area in tropical forest, do not mention darts reinforced with lithic or bone points. The only documented composite darts we have found are Malayi darts from Southern India, with a detachable conical barbed steel head, linked to a short shaft by a long and thin cord wound closely around it, which were used for shooting fishes at close range [114]. The Cherokee blowgun can deliver a dart made from a very light wood or bamboo splinter at nearly twice the speed of an arrow from a native self-bow [115]. Its effective hunting range is between 12 and 18 m [116,117]. The darts of the Jahai people in Malaysia, among the most ancient blowgun hunters, weight less than one gram and can be shot precisely at 50 m [118]. On such needle-like projectiles, the micropoints from Obi-Rakhmat would be disproportionate in every respect (width, fitting, weight distribution, inertia, ballistic properties) out of proportion.

In the context of the Obi-Rakhmat lithic industry, regarding the micropoints as arrow heads is the working hypothesis most compatible with the fundamental principles of hunting weapon design and with the available experimental references. The most disconcerting element of this provisional conclusion is the age of the levels from which these points originate. With what can we compare them?

## Comparison with former, contemporary or similar projectile heads

The earliest lithic weapon armatures reported in the literature on a traceological basis are those from Kathu Pan 1 in South Africa, stratum 4a, which has yielded the oldest dated Fauresmith Industry [119–122], at the transition between the Early and Middle Stone Age, ca 500 ka. This KP1 assemblage contains numerous unifacially retouched points and unretouched triangular flakes and blades from Levallois knapping. In a sample of 210 points and fragments covering 4 square units, 31 *diagnostic impact fractures* (DIF) were detected on 29. However, only 5 photographs of these DIF are provided, including 2 tiny burin removals, but no view of the entire artefacts (photo nor drawing). These possible spearheads are not said to differ in size or morphology from the rest of the corpus. The average length of all points (retouched or unretouched, symmetrical or not) is over 7 cm. No human remains have been found with this ancient expression of Fauresmith-type industry.

Slightly less ancient (> 279 ka) are the obsidian points from the Gademotta site complex in Ethiopia, which have been identified as projectile points based on velocity-dependent microfracture features, diagnostic damage patterns, and artifact shape [36], but the interpretation has been controversial, with the apical removals seen as resulting from a sharpening process by lateral tranchet blow removal rather than projectile impact [37,123].

The lithic industry of the Misliya cave terrace, in Israel, associated with the oldest remains of archaic Sapiens found outside Africa (ca. 180 ka), consists mainly of points of different types (unretouched Levallois, retouched Levallois and Abu-Sif and Misliya points). Three traceological analyses were carried out on samples of the lithic assemblage containing the different types of points, i.e., a total of 344 points out of a total sample of 445 pieces. The aim of the first study was to find weapon heads [73], while the next 2 were exhaustive [45,124]. In total, 48 weapon heads were identified based on their microscopic damage and microscopic linear impact traces (5% of the sample). The dominant type are the unretouched Levallois points with 26 specimens. Their average length is 64 mm and their average weight is 19 grams. Nevertheless, the complete study shows that the points at Misliya are multifunctional tools also related to the acquisition of vegetal foods as well as performing craft-related activities.

Only slightly more recent are the 3 non-tanged projectile points identified in the lower section of the stratigraphy of the Ifri n'Ammar rock shelter in Morocco, dated between 143 and 171 ka [125]. They precede the tanged Aterian points of the

upper section, dated between 171–83 ka, of which 11 out of 42 have also been identified as spear tip based on their damage [126]. The way they were hafted and their size do not seem to distinguish them from common tools. They are more massive and robust than the micropoints of Obi-Rakhmat.

On the other side of the Mediterranean, further north, in the world of the Neanderthals, at the Bouheben site in France, in an assemblage of 125 Mousterian points from around the same time period (geological assignment to MIS 6) as Misliya, 6 specimens with impact scars interpreted as weapon heads are mentioned [127]. Four rather convincing macro photographs are published, but no views of the whole specimens. This suggests that there was nothing morphologically distinctive that was worth showing.

In the rich Levallois Levallois industry from level IIA of Biache-Saint-Vaast, northern France, which is slightly older (MIS 7), 16 spear points and 20 butchering knives were identified among the Mousterian points and convergent scrapers by a macroscopic and microscopic traceological analysis of 157 convergent pieces, side scrapers and Levallois implements [128]. The weapon heads are among the most elongated and symmetrical samples in the corpus.

The macroscopic analysis of 119 unretouched Levallois points from the slightly younger assemblage of Therdonne (France, MIS 7/6) has been less successful, yielding only 2 possible weapon heads but 17 butchering knives [129,130].

Although a hundred thousand years more recent, the assemblages from Angé and Bettencourt-Saint-Ouen, in the northern half of France, do not show a different pattern, with triangular-shaped pieces serving a variety of needs, among which weapon head is one of the potential uses. At Angé, lithic points were mainly produced by a convergent unipolar production scheme, but also by bipolar and centripetal Levallois schemes [131]. One weapon point was noted for its lateral damage [132], however macroscopic and microscopic analysis of 33 other Mousterian points of different sizes and morphologies revealed only harvesting and plant processing tools (Plisson, unpublished). At Bettencourt-Saint-Ouen, where points were also produced by different schemes, the analysis of 49 Levallois points revealed 1 weapon point with lateral scars, 8 butchering knives and 2 wood knives [Caspar in 133]. These two sites are contemporary with Obi-Rakhmat's level 21, but have nothing in common with its micro projectile points.

To find small projectiles heads, it is necessary to leave Neanderthals, cross back the Mediterranean Sea and go as far as the southern end of Africa. Between > 77ka and 64ka, in 3 different cultural assemblages, the Sibudu rock shelter, in the province of KwaZulu-Natal, displays a range of types and sizes of projectile armatures, from Pre Still Bay serrated bifacial lithic points to Howiesons Ports microlith quartz segments and bone points, some of which being described as arrowheads [127,134–145]. Lithic segments and bone points continued to be used locally as arrowheads well into historic times [146–148].

However, the very different lithic head designs (Levallois and pseudo Levallois *vs* serrated bifacial, foliated or segments), the dates, the distance and the absence of geographical relays prevent us from seeing any relationship between the earliest levels of Obi-Rakhmat and the different cultural layers of Sibudu. The contribution of South Africa to the expansion of Anatomically Modern Human (AMH) is more likely to have been in the direction of Asia, according to the coastally oriented dispersal model, due in part to the similarity of the Howiesons-type segments with those of the first microlithic assemblages in India and Sri Lanka [149].

The closest technical comparison is not in Africa, but in the Rhône valley in France, in a brief Neronian occupation of the Mandrin cave 54,000 years ago (51.7–56.8 interval). This cultural layer E yielded micropoints identical to those found at Obi-Rakhmat, also impacted by use as projectile heads (Fig 17). Due to their tiny size, they are interpreted as arrowheads [151]. A Homo Sapiens deciduous tooth has been found in the same layer [152]. A subsequent local Mousterian layer, with Neanderthal mandibular remains, closes the Middle Palaeolithic stratigraphic sequence at the site [153]. The Neronian with its small arrowheads is regarded as belonging to a first pioneering wave of AMH incursion into southern Europe [154]. Other sites with lithic industries using the same reduction strategy and characterized by the combination of axial-macro and micropoints/blades/bladelets are beginning to be documented at the end of the Middle Palaeolithic in Spain with the Arlanzian of Cueva Millán [155] and in Italy at Riparo l'Oscurusciuto [156]. At Cueva Millán micropoints

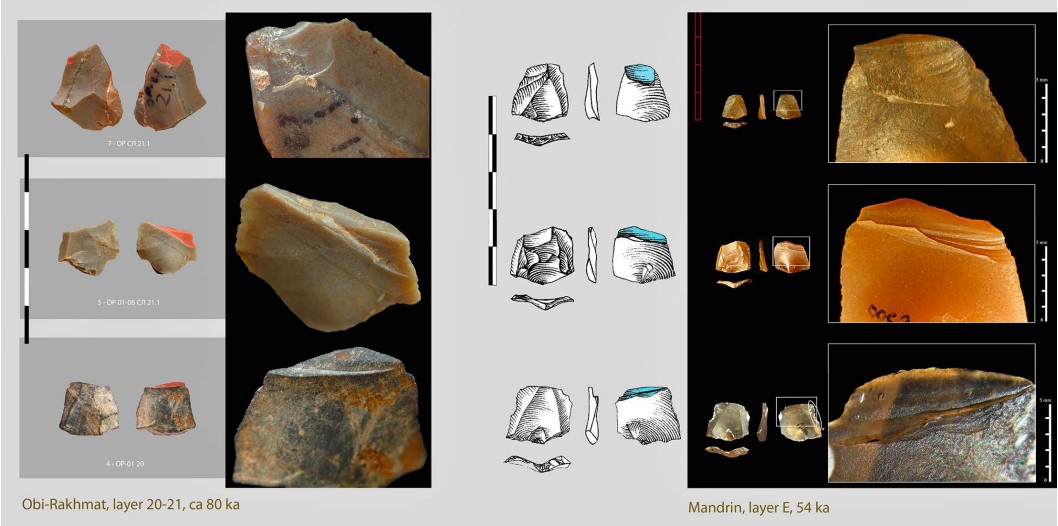

Obi-Rakhmat, layer 20-21, ca 80 ka

Mandrin, layer E, 54 ka

**Fig 17. Comparison between micropoints from Obi-Rakhmat, layers 20-21, and Mandrin, layer E [6,150].**

impacted by their potential use as projectile head are mentioned, while a past study at Riparo l'Oscurusciuto highlighted the presence of impact scars on 6 macro-points, suggesting their use as spear tips [157]. The biological identity of these innovative hunters is unknown, however, the combination of projectile heads of different sizes, including microlithic ones, does not fall within the Neanderthal repertoire [158]. The resemblance with the Neronian suggests that the same author could be involved.

## The IUP perspective

A parallel has been drawn between the industries of layer E at Mandrin and layer XXIV at Ksar Akil in Lebanon, which share notable technical similarities [152]. Due to the early excavation conditions, smallest micropoints are missing at Ksar Akil, while the dating indicates a gap of almost 10,000 years between the two assemblages. This gap is marked in the stratigraphy of Ksar Akil by an abrupt break between layers XXVI and XXIV, without any possibility of technological continuity between the Middle Palaeolithic and the Initial Upper Palaeolithic [154] and by a partial maxilla of young AMH [159]. The absence of AMH specimens in the Levant since the material from Skhul/Qafzeh (~120–90 ka ago), with the exception of the calvaria from Manot Cave in Galilee dated to more than 54.7±5.5 ka [160, but see 161], does not support a continuous representation and local evolution of AMHs in the Levant. The fact that the earliest dates for the IUP layers of Ksar Akil post-date the Neanderthal era could might suggest that earlier settlements of AMH returning to the Levant did not take place on sites that were occupied by Neanderthals at the time. More generally, we might ask whether recent developments in dating and calibration techniques make it possible to compare, in absolute terms, results from different contexts spread over almost a decade?

Whatever the timing of the return to the Levant of the AMHs, considered to be the artisans of the IUP, the milestones in the evolution of hunting weapons that would allow the link to be made are currently lacking in East Africa or Arabia. In the Levant itself, the projectile head design known from Umm El Tlel or Nahal at around 60 ka does not differ from the rest of the toolkit, as in any earlier Neanderthal site [23,82,158,162–164]. That is why Ludovic Slimak suggests looking further east for the possible roots of the very first pioneer groups of Sapiens to enter Europe [154].

Despite the distance in time (~ 25 ka) and space (~ 6000 km) that separates the first occupations of Obi-Rakhmat from the layer E of Mandrin, we can only be surprised by the great morphological similarity of their micro-projectile points,

which look interchangeable. As well as at Cueva Millán and Riparo l'Oscurusciuto, they are resulting from comparable flexible flaking patterns. The massive blades of Obi-Rakhmat are absent from the more recent assemblages, but all four are characterised by flat-faced and semi-rotated cores, and by the production of axial points, blades and bladelets.

We do not pretend to draw a direct link between Obi-Rakhmat and Mandrin, but these striking similarities in one element relating to a complex weapon system, of which this could be one of the earliest expressions, give rise to a number of reflections. In the same way that the technical solutions developed around 64 ka at Sibudu, in South Africa, persisted well into the Holocene, far beyond the particular cultural group that invented them, it is likely that micropoints of the type described here and what they encompass spread between different groups, as bow and arrows are particularly well suited to hunting game that is difficult to approach in an open foothill environment, such as the Asian ibex, and to broadening the spectrum of hunted fauna.

On the basis of recent genetic data the Persian Plateau, at the northeastern periphery of which Obi-Rakhmat is located, has recently been defined as a population hub where the ancestors of all present-day non Africans lived between the early phases of the Out of Africa expansion (~70–60 ka) and the broader colonisation of Eurasia (~45 ka) [165]. This resource-rich environment, with its diversity of topography and water sources [166], provided a refuge area conducive to demographic regeneration after the Out of Africa bottleneck, and consequently to groups interaction. These interactions probably encouraged technical innovations in response to the new environmental variables, including projectile weapons [165].

The visibility of the IUP in the Levant and Europe from ~ 45 ka may be the consequence of a same population spreading from the Persian Plateau [the second wave in the model proposed by Ludovic Smlimak: 154]. This could explain the morphological difference between the light projectile armatures of the Emiran/IUP sequence from Boker Tachtit [8] and those from Mandrin or Obi-Rakhmat, which would represent an initial, i.e., more archaic, phase. This is suggested by the fact that the shape of the micropoint was obtained in different ways. It was not yet the primary objective around which productions schemes were organised. Light projectile points, whose functional design imposes standardisation, were initially added on the margins of pre-existing production systems which were centred on other requirements. This corresponds to what the philosopher Gilbert Simondon [167,168] sees as the birth of a new technical lineage, whose initial stages of concretising the new principle on which it is based borrow from the technical environment in which it emerges, before tending towards the conditions of its own coherence. By becoming the structuring element in tool production due to their importance within the subsistence system, projectile points, through their specialization and standardization, probably influenced Sapiens' technological conceptions in a way that became his signature.

At this point, the question of the biological identity of Obi-Rakhmat's inhabitants can no longer be avoided. In 2003, teeth and skull fragments from a single juvenile individual (9–12 years old) were discovered in layer 16 (ca. 70 ka) expressing a relatively Neanderthal-like dentition but coupled with more ambiguous cranial anatomy [47,62,64,169,170]. The most immediate assumption would be that it was a Neanderthal- [170] or Denisovan-Sapiens hybrid, which could explain his premature death.

The only known association of Neanderthal remains with different types of weapon points, in this instance lithic and bone points, is in the Castelperronian of Arcy sur Cure cave in France. However, contamination with the underlying Mousterian level, which was rich in Neanderthal remains, and the recent identification of an anatomically modern ilium of a neonate [171], call such an association into question [172], while at Saint Cézaire, in France, the combination of Castelperronian lithic industry and human remains seems to be the result of solifluction [173]. As for knowledge of Denisovian technical systems, it is currently based on a single site where the only projectile element identified to date is synchronous with the presence of AMH [174,175].

If we consider the diversification of projectile armatures and the invention of propulsion instruments with their inherent complexity (systemic integration of a large number of elements) [176] as markers of a development specific to Sapiens, we need to reformulate the question of the transition between the Middle Palaeolithic and the Upper Palaeolithic, not in biological terms, between hominins (in fact, historiographically between Neanderthals and Sapiens, the Denisovans being

excluded despite their genetic legacies) but in relation to the steps in the trajectory of AMH, probably of a demographic nature [177]. This would give back to the word *transition* the notion of continuity that it conveys.

For now, the absence of genomic or proteomic analysis at Obi-Rakhmat leaves the question open. Whatever the answer, it will be of the utmost importance to the debate on the roots of the IUP.

## Conclusion

The results presented here are based on a preliminary study on material from the oldest layers of Obi-Rakhmat, which is currently under study. The number of pieces identified as projectile points may seem small, but it covers only 20 square metres. In any case, the sample is sufficient to highlight the presence of micropoints impacted by use as projectile heads in these layers dated back to ~ 80,000 years ago. Their almost microlithic shape makes them technically unsuitable for mounting on anything other than arrow-like shafts. The same type of armature is described in a pioneer settlement by Sapiens in the Rhône Valley 25,000 years later and similar point productions are beginning to be discovered at other sites, which renders the suggestion of childlike making anecdotal.

At Obi-Rakhmat, they are present in the inventory with more robust points and with bladelets. These different elements seem hard to match on the same shaft and probably corresponded to 3 types of weapons.

The bladelets, which are also resulting from several flaking patterns, will require a dedicated functional study to determine whether they were commonly used as projectile barbs. The next task will be to see whether the same technical solutions are present throughout the stratigraphy or whether this combination of weapons was specific to the first occupation layers. The question is especially relevant for the micropoints, which are about to become an index fossil. So far unnoticed because they are unretouched, tiny and fragmentary, it is likely that they will now begin to appear in sites between Central Asia and Western Mediterranean Sea. Should this be the case, it would be worthwhile to observe how Obi-Rakhmat fits into the dynamics of their dissemination: as a recipient site or as part of the cultural complex in which they were invented? The micropoints could be indicative of interconnections between distinct groups and of their temporality in relation to climatic phases.

The challenge of integrating typological, technological, chronological, geographical and anthropological data relating to the Initial Upper Palaeolithic and its premices into an unifying model suggests that what this concept encompass, in relation to the expansion of Sapiens into Eurasia, was not a sudden linear phenomenon. Instead, it was the result of a mosaic of interactions over millennia between groups emerging from Africa and vernacular populations [178]. Small projectile points and the complex weaponry behind can provide a discriminating criterion in time and space for getting a clearer view.

## Supporting information

**S1 Note. Additional methodological comments.**
(DOCX)

**S1 Table. Technological inventory from layers 20 and 21.**
(PDF)

**S2 Table. Experimental arrow heads in local silicified limestone: shot results.**
(PDF)

**S3 Table. Weapon heads and inserts: morphometric data.**
(PDF)

**S4 Table. Raw data for Fig 15.**
(PDF)

**S1 Fig.** **Retouched point 26 – OP 10 06 21.1 622, photographic view, centimetric scale.**
(JPG)

**S1 File.** **Retouched point 26 – OP 10 06 21.1 622, 3D model.**
(PDF)

**S2 Fig.** **Retouched point 28 – OP 11 21.2 06 КВ 145, photographic view, centimetric scale.**
(JPG)

**S2 File.** **Retouched point 28 – OP 11 21.2 06 КВ 145, 3D model.**
(PDF)

**S3 Fig.** **Apical fragment of retouched point 22 – OP 11 21.1, photographic view, centimetric scale.**
(JPG)

**S3 File.** **Apical fragment of retouched point 22 – OP 11 21.1, 3D model.**
(PDF)

**S4 Fig.** **Apical fragment of retouched point 104 – OP 01 19 4, photographic view, centimetric scale.**
(JPG)

**S4 File.** **Apical fragment of retouched point 104 – OP 01 19 4, 3D model.**
(PDF)

**S5 Fig.** **Retouched point 24 – ОР СЛ 21 1 196, photographic view, centimetric scale.**
(JPG)

**S5 File.** **Retouched point 24 – ОР СЛ 21 1 196, 3D model.**
(PDF)

**S6 Fig.** **Broken retouched point 20 – OP 10 21.1 0–6, photographic view, centimetric scale.**
(JPG)

**S6 File.** **Broken retouched point 20 – OP 10 21.1 0–6, 3D model.**
(PDF)

**S7 Fig.** **Apical fragment of retouched point 30 – OP 11 21.2 П6 17, photographic view, centimetric scale.**
(JPG)

**S7 File.** **Apical fragment of retouched point 30 – OP 11 21.2 П6 17, 3D model.**
(PDF)

**S8 Fig.** **Apical fragment of retouched point 27 – OP 11 П 7 21.1, photographic view, centimetric scale.**
(JPG)

**S8 File.** **Apical fragment of retouched point 27 – OP 11 П 7 21.1, 3D model.**
(PDF)

**S9 Fig.** **Broken micropoint 4 - OP-01 20, photographic view, centimetric scale.**
(JPG)

**S9 File.** **Broken micropoint 4 - OP-01 20, 3D model.**
(PDF)

**S10 Fig.  Broken micropoint 5 – ОР 01–08 СЛ 21 1, photographic view, centimetric scale.**
(JPG)

**S10 File.  Broken micropoint 5 – ОР 01–08 СЛ 21 1, 3D model.**
(PDF)

**S11 Fig.  Broken micropoint 7 – ОР СЛ 21.1, photographic view, centimetric scale.**
(JPG)

**S11 File.  Broken micropoint 7 – ОР СЛ 21.1, 3D model.**
(PDF)

**S12 Fig.  Broken micropoint 8 – ОР 11 П 7 21, photographic view, centimetric scale.**
(JPG)

**S12 File.  Broken micropoint 8 – ОР 11 П 7 21, 3D model.**
(PDF)

**S13 Fig.  Broken micropoint 9 – ОР 01 СЛ20.3, photographic view, centimetric scale.**
(JPG)

**S13 File.  Broken micropoint 9 – ОР 01 СЛ20.3, 3D model.**
(PDF)

**S14 Fig.  Broken micropoint 11 – ОР 2001–2008 СЛ 21.1, photographic view, centimetric scale.**
(JPG)

**S14 File.  Broken micropoint 11 – ОР 2001–2008 СЛ 21.1, 3D model.**
(PDF)

**S15 Fig.  Broken micropoint 16 – ОР 2001–2008 СЛ 21.1, photographic view, centimetric scale.**
(JPG)

**S15 File.  Broken micropoint 16 – ОР 2001–2008 СЛ 21.1, 3D model.**
(PDF)

**S16 Fig.  Broken retouched micropoint 2 – ОР 21.1 280, photographic view, centimetric scale.**
(JPG)

**S16 File.  Broken retouched micropoint 2 – ОР 21.1 280, 3D model.**
(PDF)

**S17 Fig.  Levallois micropoint 00 – ОР Сл 21 Х7, photographic view, centimetric scale.**
(JPG)

**S17 File.  Levallois micropoint 00 – ОР Сл 21 Х7, 3D model.**
(PDF)

**S18 Fig.  Broken retouched bladelet 19 – ОР 11 СЛ 21.1 П8, photographic view, centimetric scale.**
(JPG)

**S18 File.  Broken retouched bladelet 19 – ОР 11 СЛ 21.1 П8, 3D model.**
(PDF)

**S19 Fig.  Broken burnt crushed bladelet 15 – OP 11 21.1 07, photographic view, centimetric scale.**
(JPG)

**S19 File.  Broken burnt crushed bladelet 15 – OP 11 21.1 07, 3D model.**
(PDF)

**S20 Fig.  Backed bladelet 31 - OP-08 Сл 21.1 КВ M7, photographic view, centimetric scale.**
(JPG)

**S20 File.  Backed bladelet 31 - OP-08 Сл 21.1 КВ M7, 3D model.**
(PDF)

**S21 File.  Broken micropoint 6 – OP 2001–2008 СЛ 21.1, 3D model.**
(PDF)

**S21 Fig.  Experiments by Vladimir Kharevich and Alëna Kharevich, photographic view.**
(JPG)

**S1 Inclusivity.  Inclusivity in global research.**
(DOCX)

## Acknowledgments

We are grateful to the staff of the National Center of Archeology, Academy of Sciences of the Republic of Uzbekistan, for facilitating access to the collection. Special thanks to Dr. Françoise Courmelon and Dr. Colline Bataille for making possible the participation of H.P. to this study.

## Author contributions

**Conceptualization:** Hugues Plisson, Ksenya A. Kolobova.

**Data curation:** Alena V. Kharevich, Farhod A. Maksudov.

**Formal analysis:** Ksenya A. Kolobova.

**Funding acquisition:** Hugues Plisson, Andrei I. Krivoshapkin.

**Investigation:** Hugues Plisson, Alena V. Kharevich, Vladimir M. Kharevich, Lydia V. Zotkina, Malvina Baumann, Andrei I. Krivoshapkin.

**Methodology:** Hugues Plisson, Vladimir M. Kharevich.

**Project administration:** Andrei I. Krivoshapkin.

**Resources:** Hugues Plisson, Ksenya A. Kolobova, Andrei I. Krivoshapkin.

**Software:** Pavel V. Chistiakov.

**Supervision:** Andrei I. Krivoshapkin.

**Validation:** Hugues Plisson, Ksenya A. Kolobova, Andrei I. Krivoshapkin.

**Visualization:** Hugues Plisson, Pavel V. Chistiakov, Lydia V. Zotkina, Malvina Baumann, Eric Pubert, Ksenya A. Kolobova.

**Writing – original draft:** Hugues Plisson, Alena V. Kharevich.

**Writing – review & editing:** Hugues Plisson, Alena V. Kharevich, Malvina Baumann, Andrei I. Krivoshapkin.

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
