## [Decision Letter · Decision Letter 0]

11 Mar 2025

Dear Dr. Plisson,

We look forward to receiving your revised manuscript.

Kind regards,

Marco Peresani

Academic Editor

PLOS ONE

3. In your manuscript, please provide additional information regarding the specimens used in your study. Ensure that you have reported human remain specimen numbers and complete repository information, including museum name and geographic location.

For more information on PLOS ONE's requirements for paleontology and archeology research, see https://journals.plos.org/plosone/s/submission-guidelines#loc-paleontology-and-archaeology-research .

Additional Editor Comments (if provided):

Reviewers' comments:

Reviewer's Responses to Questions

**Comments to the Author**

1. Is the manuscript technically sound, and do the data support the conclusions?

Reviewer #1: No

Reviewer #2: Partly

2. Has the statistical analysis been performed appropriately and rigorously?

Reviewer #1: Yes

Reviewer #2: I Don't Know

3. Have the authors made all data underlying the findings in their manuscript fully available?

Reviewer #1: Yes

Reviewer #2: No

4. Is the manuscript presented in an intelligible fashion and written in standard English?

Reviewer #1: No

Reviewer #2: Yes

Reviewer #1: the manuscript provides new intresting date on an important site however I recomand that more effort will be invested it the way the authors choose to present it, especially the part of the methodology which needs better arrangement and reconsideration. the part on the traceology is marginal and the conclusions rely more on the morphology of the arrows and comparisons to other sites. I listed some remarks and recommandations in the file attached. there are many technical problems (I listed only a few) in figures and text.

Reviewer #2: Review for the manuscript PONE-D-25-02378 Arrow heads at Obi-Rakhmat (Uzbekistan) 80 ka ago

Summary

The manuscript under review aims at documenting a behavior unreported yet at the site of Obi-Rakhmat, Uzbekistan. The authors present this work as a preliminary study focusing on small size convergent blanks described here as Levallois micro-points. A preliminary use-wear study combined identified fractures consistent with impacts, and the small size of some of the blanks, techno-typological consideration and examples from the literature lead the author to speculate on micro-points used as arrowheads, maybe a bow or a bow gun. Furthermore, it is suggested that this behavior is most often associated with AMH and their lineage – which in turns could inform on the taxomic identity of the OBR inhabitants, or the diffusion of ‘pioneer’ Homo sapiens groups. The authors bring great background info in the introduction, and interesting observations on the breakage patterns of some of the objects. I do not think they demonstrate (yet) a use of mechanically assisted projectile, let alone connections with specific human taxa. They do provide, however, ground for a working hypothesis to be tested in ways that are more extensive. I would suggest to add an question mark in the title. Although this study offers many interesting facts and ideas, I suggest that revision is needed to add some missing data essential for understanding the logic of the argument, and to tune down some of the conclusions. In full disclosure, I am not qualified for evaluating fully use-wear analyses.

General comments

There is no doubt that the last three decades of techno-typological studies on lithic assemblages (and to a smaller extent, on other materials) show a strong interest for small size lithic blanks and tools. From the small flakes/tools in the Lower Paleolithic, to the UP bladelets, micro-Mousterian, asinipodian, and pontinian in Europe, nano-points and segments in Africa, and so on…words such as ‘miniaturization’, or ‘microlithization’ became popular or were re-branded to discuss implications of what seems to be a general evolutionary trend, accelerating during/after the MP-UP transition. One of the point of contention is that productions and uses of small objects go way back before the ‘transition’, and second, that examples vary in shape, size, frequencies, degree of standardization, and most probably, …in function. At least, that is what mere logic would indicate since there is little we know about their function.

Specific comments

- Criteria of identification: it is a clear and helpful section of the paper. However, some of the statements seem to undermine the conclusion reached. For example, p.5, line 110 (and thereafter) insist on the need for standardization. The latter is a relative notion, and in spite of the small number of objects, it would be good to provide coefficient of variation for the set, and compare it with pointed objects not consider as projectiles – to document the standardization. At first glance, these objects are poorly standardized if not isolated from larger convergent blanks, and their small size may provide a false impression of standardization following power laws. In short, document the connection between the idea of standardization and the material studied.

- The micro-points sample: the material studied here belongs to the layers 20 and 21. A selection process is briefly described, I suppose using the criteria listed for the identification of weapons described by the authors. The table provided is a technological inventory for the layer (and sub-layers) and it is helpful. However, an additional table describing the protocol of analysis should be provided in the Material and Method section is essential for critical assessments by the reader. It should present an exhaustive list (actual numbers and frequencies relative to the sample, and relative to the assemblage) of the material studied and discarded besides the material selected. Numbers occur in the description of the sample (48 weapon heads, 5% of the sample) but it is essential to add more in text reference to the sample size is needed for the reader to follow the description every time a sample is mentioned (e.g. massive points are e few (N=xxx; %=xxx)). The authors note that the sample is small but that given the surface; it may be not so small. This issue is complicated for various reasons but should not be evacuated too fast. It is indeed very small if one consider their frequencies over the total assemblage, and there should be more discussion regarding their frequencies within the ‘point’ category. Do frequencies support the idea that small points are a specific response to an actual need (eg. selective pressure)? Or is it consistent with stochastic variations within such assemblage? Consistent with both? This is hard to assess because the paper in its current form does not present this data in a clear fashion. The issue of density seems irrelevant here (it could result of structuration of space, or time-averaging) and the sample size should be scaled toward the claims that are made (including the ‘priors’, or how exceptional the behavior is, and the ’posteriors’, the significance of the finds for the field of research as a whole). The larger the claim, the larger the sample needed to support it. My suggestion is to tune down the discussion, especially the parts regarding strict associations with H. sapiens and implications for hominin dispersals. If it is a preliminary study and the sample size is small, it is premature to draw large conclusions.

- Micro-point technology: It seems essential to assess whether the idea of ‘micro-points’ as a response to specific needs is the most parsimonious (as stated in the paper), or not. To do so, one needs to invalidate alternative hypotheses (or at least show that there aren’t too many…). The natural expectation would be that specific needs, standardized objects, would come with specific production schemes, while opportunistic ones could rely on more casual recycling processes. Some additional data should be provided for the readers to make an opinion. For example, text and figures seems to indicate that a specific core morphology corresponding to this production cannot be isolated. Instead, different cores are illustrated as potential candidates for their production, based on some of the removal negatives highlighted (are they complete negatives? How was it determined?). This makes sense, since core types alone, at discard, are not always indicative of a single reduction method. If so, then how is the angle the striking platform/flaking surface angle of cores, at the stage of discard, comparable to blanks obtained during the reduction? How can truncated facetted be considered as occasional cores, given the dynamic of the reduction process? In addition, the authors should clarify that they are using Levallois as a type, not as a technology, given that some of the Levallois blanks are obtained using non-Levallois methods. Also, illustrations suggest that some of these removals may play a technical management role on the bladelet core flaking surface, potentially having a technological and/or another functional role in the overall technical system. However, how the technological analyses (listed as ‘attribute’, but which one?) reconstructed those schema is not explained in the paper. It is noted that attributes were recorded on points and cores, but what about technical pieces involved in their production? How do this material stand out/blend in with the rest of the assemblage? The idea here is to identify whether there is a recycling of objects that are ‘by-product’, or are imbedded, in other productions. One thing that seems very important here is to rule out the possibility that (some) of these convergent elements are not predetermining flakes coming from the production of predetermined larger Levallois blanks (e.g; to shape the basal triangle of larger elements).

- Statistics: I am a little confused regarding the U pair-wise comparison test proposed here. The figure does not have caption (sorry if I could not find it) and I do not understand what is presented here, and what it shows technologically. What is the null-hypothesis? What are the data input? What are the assumptions/requirements for this test? Why a ranking method? What is the boxplot illustrating? Without these explanations, it is not possible to understand the meaning of it and I suggest adding it in the main text.

- Impact fracture/use wear: I am not qualified for a detailed evaluation of the methods and results obtained here, but I note a few points that caught my attention. I note that an experimental effort has been performed to address other plausible cause of damage such has knapping accident (step fractures?). I assume that other functions have been considered too? It would be very important for the reader to present the detailed protocols of such experimentation, as far as can be. As the authors note, there is a broad range of opinions in the literature regarding what discriminate projectiles fractures from other uses and for non-specialists, it is quite confusing. Criteria listed mentioned that some morphologies could fit dart and spears, but could they fit other use? Small size prevent a hafting with binding, does this means that they were hafted anyway? In other studies, some authors have relied on a single method (TCSA/TCSP) or piled up all methods to see what comes out, at best reaching the conclusion that some fractures are consistent with projectile impact damage. Are they consistent with other damage causes? Mandrin is a study for which authors are surprisingly much more affirmative regarding the use of not only projectiles, but also on the kind of mechanically assisted type of projectiles used (bow and arrows). It seems to me, at best, controversial. I am not thinking of ideas like ‘small points are made by small people’ as more parsimonious, but more like damage, other uses than projectiles, or other kind of projectiles as being solid alternative hypotheses that have not been convincingly invalidated. Hence, it does necessarily support the current study, unless their results are taken for granted (which they are not necessarily for all specialists). It even makes the argument looks a bit circular. Wouldn’it be safer to suggest this as a hypothesis to be properly tested? If yes, I advise to reframe part of the text to make it clear that this is merely a possibility until a more extensive test is performed.

- Discussion: Although there is a very useful summary of the literature provided, the scope is very large. Hence, one cannot help thinking that based on preliminary analyses on a small sample size; it is a little premature to discuss hominins biological and cultural phylogeny. I guess there should be a way to present this more as a ground for future research, or shift part of the discussion in the introduction to raise awareness or interest on the question. In my personal experience, such small triangular blanks exist in many, many assemblages but haven’t been the subject of many studies yet. Hence, at this point, the exclusive connection with a lineage/species is based on the absence of published evidence. The positive side of this situation is that this is encouraging for future studies, before an actual pattern can be identified. Likewise, the discussion on the IUP lacks references and seems out of place. The material presented here is much older, and there is no apparent connection with the relevant assemblages. I would not advise to extend the discussion but instead, to shorten it. It seems normal to mention it among other time transgressive examples, but it is unnecessary to devote a whole section to it (why not on actual UP, since bladelets predominates?). I suggest avoiding hyperbolic conclusions; reduce some paragraphs that are misleading and reframe the discussion around the potential for further studies.

Smaller comments:

- Throughout the text: Linnean binomial taxonomy should be italicized, genus names takes capital letter, species names take lower key: Homo sapiens

**Do you want your identity to be public for this peer review?** For information about this choice, including consent withdrawal, please see our Privacy Policy

Reviewer #1: No

Reviewer #2: No

---

## [Author Response · Author response to Decision Letter 1]

30 May 2025

Dear Editor, dear reviewers,

We have addressed all the comments and are truly grateful for your help in improving the second version of our work in a number of ways. We may disagree on some points, however the text has been enriched by the elements discussed.

Editor :

To summarize the many points listed in a supplementary detailed report, R.1 thinks that the manuscript is worthy to be considered for publication pending several interventions you are recommended to do. These regards the way the manuscript should be presented and arranged, the methodology, which is chaotically presented and the conclusions, which are disharmonic because this study relies more on the morphology of the lithic points than on traceology. Reviewer 2 raises critical points in each section of the manuscript. Particularly, authors are requested to better demonstrate if the sampled items are standardized or not, also in function of the identification criteria adopted, considered the small number of objects. Calculation of the coefficient of variation should be provided and compared with other pointed objects. Other points regard the core types, the attributes used for the technological analysis and the possible multiple significance of convergent elements, other than the protocols used for experimentation. Furthermore, the discussion on the IUP lacks references and seems out of place, given the chronological position of the material presented here. Overall, the small sample and the preliminary state of this study, alternative hypotheses other than micro-points as a response to specific needs should be considered and tentatively invalidated. R. 2 recommendation to the authors is to tune down many claims in the conclusions and revolver the discussion around the hypotheses of a work in progress, especially when authors explore the relation of the human taxonomy and hominin dispersals. Authors should ground for a working hypothesis to be tested in ways that are more extensive and, for the reasons explained in the R’s report, they are suggested to add a question mark in the title. English editing is recommended too.

Our conclusions do not rely more on morphology than on traceology. These are two sides of the same coin. There are no more traceological criteria today for identifying an arrowhead tip than there are for distinguishing by their microwear a butcher’s knife from a sacrificial knife (Sigaut, 1991). It is the integration of other criteria, especially morphological ones, that allows us to move from trace to instrument. Dimensions are a key parameter in determining the functional capacity of any tool; a jeweller's hammer is not the one of a quarryman.

Even if we consider that such small points could nevertheless have been used to arm darts shot with spearthrower, this would still be the earliest evidence to date of a mechanically propelled projectile, what would hardly change the perspective of the paper. However, Ibex hunting in an open hill environment with this type of weapon was probably not the easiest thing in the world.

The alternative hypothesis that micropoints were not a response to specific needs would be that they were produced for no reason and opportunely used as weapon tips, regardless of their size, alongside other uses.

As for the final discussion, the IUP did not come out of from nowhere. Its roots probably run deep, and the moment of its visibility in Western Eurasia should not be confused with its “invention”. This is a view inherited from the history of prehistoric science, which was initially Eurocentric. We can't ignore the astonishing similarity between Mandrin's projectile microtips and those of Obi-Rakhmat on the grounds that they are too far apart in time, while huge territories between them remain largely unexplored. This type of unretouched tiny point can't be noticed if it's not looked for. Had it not been for Slimak's suggestion to look further east than the Levant (Slimak, 2023), they might have remained in the bags of lithic debris from Obi-Rakhmat. We're not saying there's a link, but the resemblance is worth discussing in the current state of the puzzle. if relevant bibliographical references are missing, we'll be happy to include them.

Reviewer 1 :

In the introduction the authors declare their goal at line 55 – “We present here the first results of a search for weapon points” and then review the methods that were previously used to do so however by reading through it a notion rises that there is no way of securely doing it. A clear declaration by the authors is needed to show how they are approaching this serious problem:

We have added the following clarification: “We present here the first results of a search for weapon heads in the oldest layers of the Obi-Rakhmat rock shelter, at around 80 ka, based on traceological and technological criteria”. Except finding a complete preserved arrow, dart or spear, there is no way for a completely secure weapon identification, however the convergence of different ranges of evidence and the recurrence of observations are generally robust approaches in archaeology. As far as the identification of weapon points is concerned, decades of use-wear analysis show that the methodological bases are well established. However, as on any other topic, there is sometime a temptation to adopt more simplistic approaches.

Lines 70-73: theoretical potentialities with the quote by Bordes – the authors start their review on how arrows can be studies with this claim. I would remove it as this collapses everything that comes later (…)

We updated this section “Weapons in focus”, but chose to keep Bordes' quote because it clearly explains why it can be hard to identify a weapon based only on TCSA/TCSP indices.

(…) Line 97 – “There are two levels of analysis in the identification of a projectile armature” – should add: “to our view” because this is not a conventionality. And: this is given after the example on the Ethiopian points – however the aspects given here can also provide a misleading result as in the case of the Ethiopian points.

Not only in our view : first question – is it a weapon head?, second question – from which weapon? In the case of the Ethiopian points, the reply to the first question was in fine negative. Traceological features are limited in number. They are the same for describing shaping and wear, but distinct combinations make it possible to write very different stories.

And then 119 – “To put it simply, the design of a point will be very different depending on whether you're hunting rabbits with a bow or aurochs with a spear” – do the authors mean here that we do not need these analytical techniques?

In this case, we wanted to say that different sizes of game require different hunting weapons. We removed this phrase from the main text to avoid misunderstandings.

153 – goat – better put ibex

Done.

Materials and methods – in general the chapter needs revision: authors jump from one subject to another and some details are missing. Explanations on the traceological aspect and compared to the technological aspect is not proportional when compared to the body of information given in the results and the discussion clearly shows that the authors do not rely on the results of the traceological study. It seems that the results are based on the traceological study however this part is applied on a limited number of artifacts and too speculative. On the other hand, the technological study is comprehensive and detailed, providing valuable information on how the points were produced. Other aspects addressed in the discussion such as the comparisson with points from different parts of the world should be mentioned in the methodology.

We sincerely appreciate this valuable comment. In response, we have thoroughly revised and expanded the "Materials and Methods" section to provide: (1)A more detailed description of the techno-typological analysis procedures; (2) A comparative analysis of our sample with points from different global regions; (3) The traceological criteria used for sample selection.

The traceological criteria, initially introduced in the "Weapon in Focus" section, have been significantly enhanced and moved to a Supplementary Note, since it is not only an enumeration of methods and equipment used, but also our viewpoint on certain aspects of the method and a summary of the criteria we consider relevant. For conciseness, the main text retains a summary of these principles with a cross-reference to the Supplementary Note.

165 – “The analysed sample.... includes typological pieces and a search for triangular shapes” – rephrase...not clear

Rewriting: “It includes typological pieces and small triangular flakes found among the bags of lithic debris from layers 20 to 21 stored at the National Center of Archeology in Tashkent, Uzbekistan. The first sorting, looking for impact damage, was done with the naked eye”

173 – Photomacrographs..and add: micrographs (because a high magnification was used). Was mag of up to 500×? I don’t think that this was used, please specify the mags used in the analysis.

Photomacrography belongs to the professional terminology (eg. Close-up photography and photomacrography, Kodak technical publication. Eastman kodak co., 1969).

Photomicrography already mentioned : “photomicrographs using a Nikon D750 on the phototube of the microscope”. Line 173.

x100, x200 and x500 magnification were used in search of MLIT on archaeological and experimental samples. x500 magnification is mentioned on the shots.

x500 is unusual magnification because most traceologists use a short working distance objective with very low depth of field ; we are using a long distance one with comfortable depth of field.

175 – Helicon Focus©

Done.

176 - The artefacts were scanned – not clear what is the purpose of this – should be explained in the methodology.

We have expanded the Methodology section to include a detailed description of our 3D scanning procedures and their purpose. The primary objective of this instrument is to enable independent evaluation of points by other researchers - to facilitate this, we have provided the complete 3D models and high-resolution photographs of each artifact in the Supplementary Materials.

178 – “The macroscopic diagnostic features.... corpus of over 500 flint points and barbs....” on page 134 the authors mention (only one time throughout the publication) that the lithics are of “silicified limestone” – this is a unique material, does it respond to impact the same way? Does MILT form or easy to observe? There is no real discussion on the lithics – fig 7 – looks like flint? Scans look like limestone? Needs further clarification.

Impact marks do not change from one rock to another, but their intensity may. Obsidian, for example, is much more fragile than basalt. This is why a small experimental series has been carried out (please, see S22 Fig). Silicified limestone reacts macroscopically in the same way as flint while MLIT are only visible on the best-crystallized variant. Different degrees of crystallization are observed in the outcrops around the site. This clarification now appears in the main text.

182 – “Additional preliminary experiments (S21 Fig)” – are these knapping experiments? The fig is a scan of a point – not clear.

S21 Fig is a plate showing different aspects of the additional experiment, while S21 File is a 3D model.

184 – “This step enabled us to observe the knapping accidents” – needs elaboration – how can knapping accident be mistaken by or mimic impact damage.

Perhaps there was some misunderstanding because in the original text, we cite references describing how impact damage may be mistaken for knapping accidents or other uses. However, these traces can be distinguished by carefully examining the diagnostic criteria for each type of damage, as outlined in the literature. The main text includes full references to these sources.

Line 84-87 : Causes other than being at the tip of a spear or arrow can lead to axial compressive stresses: knapping accident (Liu et al., 2024; Newcomer, 1976; Pargeter, 2011) (S1 - 2 Fig), certain types of shaping (Tixier et al., 1980), hard butchering, use as a chisel, accidental dropping, etc. In some cases, the distinction is easy to make at the scale of the artefact itself, in others it is less so if a range of criteria is not taken into account.

Results

199 – “Traceological inventory” – revise term, first time mentioned in results, not in methodology. This chapter also includes typological aspect – should be excluded to a specific sub chapter and better presented.

We propose to change “Traceological inventory” by “Impacted armatures”.

We chose not to address detailed point typology in this study because it is particularly complex for this chronological period. Instead, we adopted a simplified classification based exclusively on size categories: large points and micro-points. This methodological approach has now been added to the revised Methodology section.

201 – “we have selected 20 pieces....” – this should appear from the beginning – the article deals with 20 items

Added in the abstract : “Three types of projectile armature are identified from their impact marks over a selection of 20”

202 – “Most of the artefacts” – referring to the 20 selected ones? Not clear

Not refering to 20 artefacts only but to the whole assemblage. Corrected : “Most of the lithic assemblage, due to the raw material and surface alteration, is unsuitable for microscopic analysis, however MLIT have been found on two micropoints.”

205 – “which depends on the proportion of compressive and bending stress that caused the material to break by buckling or percussion” – what is buckling? Anyway should move to methodology.

In structural engineering, buckling is the sudden change in shape (deformation) of a structural component under load. Moved to the methodological section.

207 – “the 3D model – these scans do not really allow experts to evaluate the items: scans cannot allow following the orientation of scars, some scans give a very limited view of the artifact (for example – ventral face only). I recommend making a regular plate that illustrate all the 20 points – it will also provide a first wonderful documentation of the artifacts.

In response to the reviewer's suggestion, we have added a photographic plate illustrating together all the 20 points (new Fig 4). Regarding the value of 3D models, we would argue that when models are opened with Adobe Acrobat, they are oriented according to graphic standards (top view, apical end up, proximal end down), and it's possible to orient them as desired and change the lighting. The 3D models are now at millimetric scale, enabling measurements to be taken directly on screen using the virtual tools provided by Acrobat.

We have added a photographic plate illustrating together all the 20 points (new Fig 4).

209 – fig. 3 – right side annotations: “not relevant, equivocal...” not clear, not explained in the text

Added : “and their degree of relevance for the recognition of projectile points.”

217 – fig. 4 – should add a, b, c... – left one – why medium size? Is this the massive point? And are these Abu Zif points?, middle one – Levallois point?, right one – retouched bladelet? – types are not adequately defined.

The notion of massiveness was in relation to the micro-points, but not to the site's lithic industry, which includes larger modules. Here, we replace “medium sized”, which is confusing, with “large”. Typology is not the concern in this paper, so we prefer to keep the distinction by size and not by typological type. Numbering and Levallois mention added.

226 – medium or massive? Inconsistent terminology for the same object?

“Massive” replaced by “large” in most of the text.

Fig 6 – impact damage is not clear by looking at these scans, anyway their location should be specified or enlarged.

Please, see the photographs of each sample in Supporting information (S1 Fig to S21 Fig).

Fig 7 and 8 – a reference to the location of the MILT shown on the right photo should be marked on the tool. What is the orange colour on the pieces? Is this flint in fog 7?

The location of the MLIT is marked by the small orange frame in the central macro view. On the right recto-verso view, t

---

## [Editor Report · Decision Letter 1]

1 Jul 2025

Arrow heads at Obi-Rakhmat (Uzbekistan) 80 ka ago?

PONE-D-25-02378R1

Dear Dr. Plisson,

We’re pleased to inform you that your manuscript has been judged scientifically suitable for publication and will be formally accepted for publication once it meets all outstanding technical requirements.

Kind regards,

Marco Peresani

Academic Editor

PLOS ONE
---

## [Editor Report · Acceptance letter]

PONE-D-25-02378R1

PLOS ONE

Dear Dr. Plisson,

I'm pleased to inform you that your manuscript has been deemed suitable for publication in PLOS ONE. Congratulations! Your manuscript is now being handed over to our production team.

Kind regards,

on behalf of

Dr. Marco Peresani

Academic Editor

PLOS ONE